# Thermophilic Filamentous Fungus C1-Cell-Cloned SARS-CoV-2-Spike-RBD-Subunit-Vaccine Adjuvanted with Aldydrogel^®^85 Protects K18-hACE2 Mice against Lethal Virus Challenge

**DOI:** 10.3390/vaccines10122119

**Published:** 2022-12-11

**Authors:** Ram Nechooshtan, Sharon Ehrlich, Marika Vitikainen, Arik Makovitzki, Eyal Dor, Hadar Marcus, Idan Hefetz, Shani Pitel, Marilyn Wiebe, Anne Huuskonen, Lilach Cherry, Edith Lupu, Yehuda Sapir, Tzvi Holtzman, Moshe Aftalion, David Gur, Hadas Tamir, Yfat Yahalom-Ronen, Yuval Ramot, Noam Kronfeld, David Zarling, Anne Vallerga, Ronen Tchelet, Abraham Nyska, Markku Saloheimo, Mark Emalfarb, Yakir Ophir

**Affiliations:** 1Department of Biotechnology, Israel Institute for Biological Research, Ness-Ziona 7410001, Israel; 2VTT Technical Research Centre of Finland Ltd., 02150 Espoo, Finland; 3Department of Biochemistry and Molecular Genetics, Israel Institute for Biological Research, Ness-Ziona 7410001, Israel; 4Department of Infectious Diseases, Israel Institute of Biological Research (IIBR), Ness-Ziona 7410001, Israel; 5Faculty of Medicine, Hebrew University of Jerusalem, Jerusalem 9112102, Israel; 6Department of Dermatology, Hadassah Medical Center, Jerusalem 9112102, Israel; 7Envigo CRS Israel Limited, Ness-Ziona 7414001, Israel; 8Dyadic International, Inc., Jupiter, FL 33477-5094, USA; 9Dyadic Netherland B.V., Nieuwe Kanaal 7-S, 6709 PA Wageningen, The Netherlands; 10Department of Pathology, Sackler Faculty of Medicine, Tel Aviv University, Tel Aviv 6997801, Israel; 11Department of Chemical Engineering, University of California, Davis, CA 95616, USA

**Keywords:** COVID-19, antigen, pandemic, SARS-CoV-2, receptor binding domain, RBD, vaccine, biomanufacturing, dyadic, zoonotic, variants of concern, *Thermothelomyces heterothallica*, *Myceliophthora thermophila*, CHO-cell, insect cells, baculovirus, neutralizing antibodies, recombinant protein subunit vaccine, toxicology, glycoprotein, glycosylation, glycans, efficacy, stability, adjuvant, alum, aluminum-based vaccine adjuvants, C-tag, BALB/c mice, K18-hACE-2 mice, intranasal challenge, virus, IgG, IgG1 and IgG2b

## Abstract

SARS-CoV-2 is evolving with increased transmission, host range, pathogenicity, and virulence. The original and mutant viruses escape host innate (Interferon) immunity and adaptive (Antibody) immunity, emphasizing unmet needs for high-yield, commercial-scale manufacturing to produce inexpensive vaccines/boosters for global/equitable distribution. We developed DYAI-100A85, a SARS-CoV-2 spike receptor binding domain (RBD) subunit antigen vaccine expressed in genetically modified thermophilic filamentous fungus, *Thermothelomyces heterothallica* C1, and secreted at high levels into fermentation medium. The RBD-C-tag antigen strongly binds ACE2 receptors in vitro. Alhydrogel^®^‘85’-adjuvanted RDB-C-tag-based vaccine candidate (DYAI-100A85) demonstrates strong immunogenicity, and antiviral efficacy, including in vivo protection against lethal intranasal SARS-CoV-2 (D614G) challenge in human ACE2-transgenic mice. No loss of body weight or adverse events occurred. DYAI-100A85 also demonstrates excellent safety profile in repeat-dose GLP toxicity study. In summary, subcutaneous prime/boost DYAI-100A85 inoculation induces high titers of RBD-specific neutralizing antibodies and protection of hACE2-transgenic mice against lethal challenge with SARS-CoV-2. Given its demonstrated safety, efficacy, and low production cost, vaccine candidate DYAI-100 received regulatory approval to initiate a Phase 1 clinical trial to demonstrate its safety and efficacy in humans.

## 1. Introduction

The COVID-19 pandemic caused by severe acute respiratory syndrome CoronaVirus-2 (SARS-CoV-2), with subsequent emergence and spread of SARS-CoV-2 variants of concern (VOC, i.e., Alpha, Beta, Gamma, Delta, Omicron), continues to present an unprecedented global health crisis [1,2]. The Omicron VOC, as compared to Delta, more rapidly replicates in human nasal epithelial tissues and, importantly, shows an extended angiotensin converting enzyme 2 (ACE2) binding tropism over other VOCs, suggesting an increased ability to infect wild and domestic animals in possible reverse zoonotic events [3]. Omicron enters cells more readily and efficiently via both membrane fusion, as well as receptor-mediated endocytosis (RME), and remarkably avoids the usual cell’s endosomal restriction of Beta, Gamma, Delta, and other VOCs [3]. Thus, to bring the pandemic under control, a widely available, equitably distributed, inexpensive, and rapidly modifiable vaccine platform is urgently needed. Such subunit protein vaccine manufacturing capabilities, having thermo-stability for ease of distribution, as well as global immunization strategies are essential against emerging zoonotic VOCs and potential reverse or forward zoonoses, especially from newly emergent mutants like Omicron, which can efficiently bind ACE2 receptors of other animals [2,3,4].

GAVI, the Vaccine Alliance, that reported the urgent need for COVID-19 vaccines has also put critical stresses on vaccine production needs for other infectious diseases (https://www.gavi.org (accessed on 20 August 2022)). Global competition for specialized reagents, supplies, manufacturing, purification, testing, cold-chain storage, and materials to increase global production capacities have resulted in the inequitable distribution of SARS-CoV-2 vaccines and other essential non-COVID-19 vaccine products. 

As of January 2022, less than half the world’s population has received even a first dose of COVID-19 vaccine, while a significant percentage of wealthier nations are administering second, third, or even fourth inoculations. It was expected that once demand fell in higher-income countries, lower-income countries would receive increased vaccine supplies. However, most high- and middle-income countries are now administering boosters and are also stockpiling doses, further contributing to a lack of equitable and affordable access to vaccines around the world [5]. 

Global veterinary and human vaccine development strategies would ideally implement rapidly scalable manufacturing methods, employing readily available reagents, enabling the rapid production of inexpensive, thermostable, mutant cross-protective vaccines. Processes that do not require extensive and expensive, labor-intensive manufacturing, purification, or stringent low-temperature storage and transport conditions, as well as costly or limited raw materials, would all be most beneficial and could be more equitably and rapidly distributed. 

COVID-19 vaccines and vaccine candidates induce anti-spike (S) receptor-binding domain (RBD) antibodies that block interactions with ACE2, the receptor for SARS-CoV-2 virus-binding and neutralize infectivity [6,7,8,9,10,11]. Specific mutations in the spike RBD may significantly increase ACE2 binding affinity while decreasing neutralizing antibody binding [4,12,13,14]. RBD is, thus, considered critically important for clinically relevant mutations in terms of potentially increased transmissibility, host range, host immune evasion, tissue and organ tropism, pathogenicity, and relative virulence [2,15,16,17,18]. 

The RBD antigenic determinant of the SARS-CoV-2 virus S protein can induce safe, effective, and protective neutralizing antibodies [4,8,19,20,21,22,23,24]. Since it is a relatively small portion of the overall spike glycoprotein, the RBD is much less difficult to produce, as it is 18 times smaller than the full-spike-trimer, six times smaller than the full spike-monomer and may be manufactured at much higher yields when compared to the entire spike glycoprotein [25,26]. In addition, purified spike RBD glycoprotein subunit vaccines do not elicit host immune responses to viral vectors (i.e., adenovirus or pseudo-virus vaccines) and pose a reduced risk of antibody-dependent enhancement (ADE) of infection or enhanced respiratory disease (ERD) [27,28,29]. 

Various different SARS-CoV-2 vaccines have received emergency use authorization (EUA) or are in pre-clinical and clinical development. 

mRNA-encoded vaccines (Moderna; Pfizer; others) adenovirus-based, non-replicating DNA-encoded vaccines (Jansen; AstraZeneca; Gamaleya; CanSino; others), inactivated whole virus protein-encoded vaccines (SinoPharm; others).

As of the end of 2022, 242 vaccine candidates were in clinical development, with 77 of these being recombinant protein subunit-based vaccine candidates. There are 50 vaccines (including licensed vaccines and vaccines under emergency and conditional use) authorized by at least one national regulatory authority [30] The various SARS-CoV-2 recombinant protein subunit vaccines include active components, such as spike-glycoprotein-monomer [26], spike-glycoprotein-trimer or RBD-glycoprotein-subunit-antigen [2,6,31,32,33,34,35,36,37]. Ten different recombinant RBD glycoprotein-subunit-antigen SARS-CoV-2 vaccines received a total of 21 emergency use authorizations (EUAs) by at least one national regulatory agency outside of the USA [30]. The protein subunit SARS-CoV-2 vaccines currently authorized for use are summarized in Table 1. All these protein subunit vaccines induce anti-RBD neutralizing antibodies with acceptable safety.

Key advantages of protein-encoded vaccines over mRNA-encoded and adenovirus-vectored or other-DNA-encoded vaccines are that they are significantly easier to produce and transport and, consequently, more affordable [5]. Certain vaccine candidates based on the spike-S1-RBD antigen generally require significantly less-stringent temperature and storage requirements for distribution and currently are more readily amenable to large-scale manufacturing [28,38,39]. A comparison of full-length S-protein to subunit RBD vaccines has demonstrated that RBD-based vaccines induce comparable humoral antibody and T-cell-mediated adaptive host immune responses against both the original Wuhan SARS-CoV-2 isolate, as well as emerging VOCs. 

In terms of their production platforms, RBD antigens are produced in a variety of bio-manufacturing systems that include: Chinese hamster ovary (CHO) cells, insect (*Spodoptera fugiperda*, *Sf9*) cells, yeast (*Pichia*, *Saccharomyces*) cells, or human cell line manufacturing systems (transiently transfected Expi293-cells and HEK 293 stable human cell lines), where vaccine manufacturing processes are very labor-intensive and/or expensive, generally slow and produce relatively low yields of vaccine antigen, which requires refrigeration. For example, the full-spike or spike-S1-RBD subunit antigen production yields for *Pichia pastoris* range from 21 to 280 mg/L [28,40,41,42,43,44,45,46,47]. The RBD-tandem repeat dimer antigen expression level in CHO cells is 1500 mg/L, with a final antigen yield of 670 mg/L of purified tandem repeat antigen [27,48]. Antigen yields for the S1-RBD antigens in HEK-293T human cells are 5–15 mg/L [40,48], and antigen yields for Baculovirus-vectored (BEVS) S1-RBD antigen in insect cells are 30 mg/L for the RBD subunit antigen [26,27] or 1–10 mg/L for the full S1 antigen [26,49,50,51]. Neutralizing antibodies are induced in vivo with full-spike, or spike-S1-RBD subunit protein vaccines manufactured in insect, yeast, hamster, or human cells, as shown in both preclinical animal and human clinical trials [6,27,34,35,48,50,51]. The recently EUA-approved CHO-cell-produced dimeric tandem-repeat RBD vaccine reached a final yield of 670 mg/L of purified antigen [48]. However, CHO-cell production of vaccine antigens is relatively slow and expensive and requires expensive media, equipment, and facilities, either not readily available or much too costly for middle or especially low-income countries.

Here we present the production of S1-RBD antigen in a proprietary genetically modified thermophilic filamentous fungus C1 clonal cell line that was specifically developed for recombinant protein manufacturing and is compatible with both local portable and global mass production scale-up manufacturing. These *Thermothelomyces heterothallica* (formerly known as *Myceliophthora thermophila*) C1 fungal cell clones are optimized for glycoprotein production [52,53]. In the present study, we report the cloning, expression, C-tag affinity purification, and testing of RBD (amino acids 333–527) of the original SARS-CoV-2 D614G Wuhan-Hu-1 strain. The results presented here indicate that DYAI-100A85 elicits safe, effective, and protective immune responses in hACE2-transgenic C57BL/6J mice against challenge with a lethal intranasal dose of SARS-CoV-2 (original D614G Wuhan-Hu-1 strain), without loss of body weight, toxicities, or adverse events. Results of the repeated–dose GLP toxicity study conducted with DYAI-100 showed that no local or systemic toxicity was observed in the vaccinated rabbits [54]. The SARS-CoV-2 C1-RBD vaccine candidate demonstrated an excellent safety profile and a lasting immunogenic response against the receptor-binding domain [54]. At the end of 2022, the vaccine candidate DYAI-100 received regulatory approval to initiate Phase 1 clinical trial to demonstrate its safety and efficacy in humans [55].

## 2. Materials and Methods

### 2.1. Construction of Spike-S1^333–527^RBD-C-Tag Production Strains

A DNA sequence coding for C1 endogenous CBH1 signal sequence, [UniProtKB-P62694 -(GUX1_HYPJE)], cbh1 gene of *Trichoderma* reesei, residues 333-527 of the spike (S1) glycoprotein from SARS-CoV-2 spike S1, strain Wuhan-Hu-1, (GenBank No.; QHD43416.1), a gly-ser-linker dipeptide and the carboxy-terminal tetrapeptide C-tag (E-P-E-A) flanked by homologous recombination sequences to the C1-cell DNA expression vector and MssI restriction enzyme sites was designed and synthesized by GenScript (Piscataway, NJ, USA). The codon usage was optimized for expression in *Thermothelomyces heterothallica.* This recombinant DNA fragment was amplified by PCR from the GenScript plasmid and cloned by Gibson Assembly (NEBuilder^®^ HiFi DNA Assembly Cloning Kit, New England Biolabs, MA, USA) method into the PacI restriction site of the C1-cell expression vector plasmid, pMYT1055 under the endogenous C1*bgl8* DNA promoter. The sequence of the recombinant plasmid construct was confirmed by DNA sequencing. 

The C1 producing strain construction was done as described in Figure 1. The expression-ready plasmid DNA vector, pMYT1142 and another vector, pMYT1140, needed for a completion of the marker cassette (*nia1* and hygromycin resistance), were digested with MssI and the two plasmids were co-transformed into the DNL155 strain, from which fourteen different protease genes were deleted. DNA transformations were performed by standard protoplast/PEG methods [56], and transformants were selected for *nia1*^+^ phenotype and hygromycin resistance. Transformants were streaked onto selective medium plates and inoculated into liquid cultures in 24-well plates from the streaks. The growth medium components contained (in g/L): glucose, 5; yeast extract, 1; (NH_4_)_2_SO_4_, 4.6; MgSO_4_·7H_2_O, 0.49; KH_2_PO_4_, 7.48; (and in mg/L): EDTA, 45; ZnSO_4_·7H_2_O, 19.8; MnSO_4_·4H_2_O, 3.87; CoCl_2·_6H_2_O, 1.44; CuSO_4_·5H_2_O, 1.44; Na_2_MoO_4_·2H_2_O, 1.35; FeSO_4_·7H_2_O, 4.5; H_3_BO_4_, 9.9; D-biotin, 0.004; 50 U/mL, penicillin; and 0.05 mg, streptomycin. 24-well plates were incubated at 35 °C with shaking at 800 RPM for four days. Cell culture supernatants were collected, and proteins analysed by western immunoblotting performed with the primary antibodies spike-S1-anti-RBD IgG, rabbit polyclonal antiserum (SinoBiologicals, Wayne, PA, USA. Cat. No. 40592-T62), and capture select biotin anti-C-tag conjugate (Thermo Fisher, MA, USA). The secondary antibodies were either goat anti-rabbit (GAR)-IgG-conjugated with IRDye-680RD (Li-Cor) or GAR-IgG-conjugated with IRDye-800CW-streptavidin (Li-Cor). The transformed clones producing the desired RBD-C-tag protein were purified from single colonies and DNA was analysed by PCR to confirm the correct integration of the transgene expression cassette and by qPCR for clone purity. 

### 2.2. PCR DNA Analysis of Recombinant Cell Clones Showed the Expected Correct RBD Expression Cassette Integration and Clone Purity

DNA-transformed cell clones producing RBD-C-tag protein were purified by single colony plating, and single colonies were analysed by PCR to determine the correct chromosomal DNA integration of the recombinant DNA expression cassette and by qPCR for clone purity. Correct DNA integration in *bgl8* locus was verified by two PCR reactions. To demonstrate the correct 5′-DNA integration, a DNA fragment of 2.6 Kb was amplified with PCR primers: oMYT0746_bgl8_5int_1 and oMYT2823_RBD_start_rev from the positive DNA transformants (see Appendix A). 

To demonstrate correct 3′-DNA integration, a fragment of 1.7 Kb was amplified with PCR primers: oMYT1277_trpCtermF and oMYT0748_bgl8_3int_1 (see Appendix A) from the positive DNA transformants. C1-cell clone purity, in terms of total loss of *bgl8* was verified by DNA amplification with qPCR primers: oMYT0529_bgl_qPCR and oMYT0532_bgl_qPCR (see Appendix A) and using C1-cell actin gene as the reference gene. The DNA transformed cell clones were analysed by qPCR as duplicate samples and no DNA fragment from the *bgl8* gene was amplified in these clones during forty cycles of amplification, whereas the reference Actin gene DNA was typically amplified in PCR cycles 21–22 with the Actin DNA primers (see Appendix A).

After PCR verification, a single transformant was selected as the production strain and stored as a master cell bank (MCB) at −80 °C. For final verification, the production strain was streaked on a selective plate and the RBD-C-tag DNA cassette within this clone was amplified by PCR and sequenced. 

### 2.3. Fungal C1 Fermentation Recombinant Subunit Antigen Manufacturing Process Development in Low-Cost Glucose Fermentation Media

The medium used for growth of pre-cultures in shake flasks contained: (g/L) glucose, 16; yeast extract, 4; (NH_4_)_2_SO_4_, 7.5; MgSO_4_·7H_2_O, 0.45; KH_2_PO_4_, 17.25; Na_2_HPO_4_·12H_2_O, 12; NaH_2_PO_4_, 6; (mg/L) EDTA, 45; ZnSO_4_·7H_2_O, 19.8; MnSO_4_·4H_2_O, 3.87; CoCl_2_·6H_2_O, 1.44; CuSO_4_·5H_2_O, 1.44; Na_2_MoO_4_·2H_2_O, 1.35; FeSO_4_·7H_2_O, 4.5; H_3_BO_4_, 9.9; D-biotin, 0.15; and Thiamine-HCl, 1.2. The batch bioreactor medium contained these same components, but with lower concentrations of glucose, phosphate and trace elements and with higher concentration of yeast extract(g/L): glucose, 10; yeast extract, 30; (NH_4_)_2_SO_4_, 6.3; KH_2_PO_4_, 0.47; MgSO_4_·7H_2_O, 0.09; (mg/L) EDTA, 0.4; ZnSO_4_·7H_2_O, 4.1; MnSO_4_·4H_2_O, 0.81; CoCl_2_·6H_2_O, 0.3; CuSO_4_·5H_2_O, 0.3; Na_2_MoO_4_·2H_2_O, 0.28; FeSO_4_·7H_2_O, 0.94; H_3_BO_4_, 2.1; D-biotin 0.03; and thiamine-HCl, 0.25. The medium for feeding the batch fungal cell cultures contained, (g/L): glucose, 500; yeast extract, 15; (NH_4_)_2_SO_4_,12.5; KH_2_PO_4_, 3.75; MgSO_4_·7H_2_O, 0.75; (mg/L) EDTA, 75; ZnSO_4_·7H_2_O, 33; MnSO_4_·4H_2_O, 6.45; CoCl_2_·6H_2_O, 2.4; CuSO_4_·5H_2_O, 2.4; Na_2_MoO_4_·2H_2_O, 2.25; FeSO_4_·7H_2_O, 7.5; H_3_BO_4_, 16.5; D-biotin 0.25; and thiamine-HCl, 2.

#### Off-Line Analytics of the Fermentations

For quantitative analytic determinations, Mycelia were collected by filtration through Whatman GF/B filters under vacuum and washed twice with an equal or greater volume of reverse osmosis purified H_2_O. Alternatively, mycelia were collected by centrifugation at 13,000× *g* in 2 mL microcentrifuge tubes and washed twice with an equal volume of reverse osmosis H_2_O. Mycelia were dried at 105 °C to a constant weight. Glucose concentrations were determined by HPLC using a Fast Acid Analysis Column (100 mm × 7.8 mm, BioRad Laboratories, Hercules, CA, USA) linked to an Aminex HPX-87H organic acid analysis column (300 mm × 7.8 mm, Bio-Rad Laboratories, Hercules, CA, USA) with 5 mM H_2_SO_4_ as eluent and a flow rate of either 0.3 or 0.5 mL min^−1^. The column was maintained at 55 °C. Peaks were detected using a Waters 410 differential refractometer and a Waters 2487 dual wavelength UV (210 nm) detector.

### 2.4. Production of the RBD-C-Tag in 10-Liter Fermenters

For fermentation initiation, Petri dishes which had been inoculated with frozen mycelium were incubated at 37 °C for 2 days. Next, 5 mL of complete medium (see below) were added to the plate and the mycelium were scraped using a sterile cotton swab. The medium was then transferred to a 250 mL shake flask containing 45 mL of complete medium, incubated at 37 °C, 250 rpm, for a further 2 days. Next, 25 mL were transferred into each of two 3.5 L conical baffled Shake flasks containing 300 mL complete medium: yeast extract, 5 g/L; (NH_4_)_2_SO_4_, 4.62 g/L; NaCl, 0.41 g/L; KH2PO4, 7.48 g/L; Casamino acids, 1 g/L after autoclaving 20 mL of glucose 50%; 2 mL of 1 M MgSO_4_; 1 mL, 1000× trace elements; and 1 mL, Spectinomycin, 150 mg/mL. These flask cultures were incubated at 37 °C, 250 rpm for 24 h.

RBD-C-tag was manufactured in a 10 L Bioflo 120 Fermenter (Eppendorf, Hamburg, Germany), and 5 L of the high concentration yeast extract medium (glucose, 10 g/L; Yeast extract, 30 g/L; 20× stock salt solution, 6.25 mL/L (as described above for the Ambr250 fermentation), (NH_4_)_2_SO_4_, 3 g/L, P2000 antifoam, 1.25 mL/L; Spectinomycin, 150 µg/L) was inoculated with a seed culture of 500 mL of the C1 production strain, grown in two different shake flasks overnight. Fermentation conditions were as follows: 37 °C, a pH of 7.0 was maintained by NH_4_OH, 12.5% and HCl, 2.5 N with 20% dissolved oxygen kept by agitation control at 200–650 rpm until glucose was depleted. When oxygen demand started to fall, a feed medium (containing glucose, 500 g/L; yeast extract, 15 g/L; 20× salt stock solution 50 mL/L, Spectinomycin 150 µg/L) was added at a rate of 16 mL/h. The 20× salt stock was as follows: (NH_4_)_2_SO_4_ 250 g/L; MgSO_4_·7H_2_O, 15 g/L; Biotin, 5 mg/L; KH_2_PO_4_ 75 g/L; Thiamine 40 mg/L, 1000× trace elements 30 mL/L (as described above for the Ambr250 fermentation). 

### 2.5. Purification of S1^333–527^RBD-C-Tag Glycoprotein Antigen Using Capture-Select™ C-Tag Affinity Matrix

Supernatants from 90 h. of fermentation of C1 production strain were centrifuged at 32,000× *g*, 10 min, 4 °C, two times, using a Sorvall RC-5C centrifuge, and filtered through a Sartopore 0.2 µm filter capsule (Sartorius, Gottingen, Germany). The clarified fermentation fluid was concentrated three-fold and the buffer was exchanged with PBS, pH 7.2 (Biological Industries, Bet haemek, Israel) using a Sartocon^®^ Slice Disposable Crossflow cassette with a 30 kDa. membrane (Sartorius). For rapid purification, CaptureSelect™ C-Tag Affinity Matrix (Thermo Fisher Scientific, Waltham, MA, USA) was used to combine the unique selectivity for the 4-amino acid peptide tag (E-P-E-A). Mild elution conditions at neutral pH were applied using magnesium chloride or propylene glycol, which ensured high recoveries of pH-sensitive target protein product(s), including RBD. The ultra-filtrated culture supernatant containing RBD-C-tag was added to 1 mL resin and gently mixed for 30 min and packed in a Poly-Prep^®^ Chromatography column (Bio-Rad, Laboratories, CA, USA) using gravity. The column was washed with 10 Column Volumes of PBS, pH 7.2. The RBD-C-tag was eluted with three column volumes of buffer containing the final concentrations of 20 mM Tris-HCl, pH 7.0 and 2 M MgCl_2_. 

Larger scale purifications were performed utilizing the ÄKTA explorer FPLC system (GE Healthcare, Buckinghamshire, UK) using a CaptureSelect™ C-tagXL pre-packed 5 mL column (Thermo Fisher Scientific, Waltham, MA, USA). These C-tag columns were conditioned with PBS pH 7.2 before a 22 mL sample application at 1 mL/min, followed by washing with 36 mL PBS, pH 7.2 at a rate of 2 mL/min, and protein was eluted at a rate of 2 mL/min with 28 mL of 20 mM Tris–HCl, 2 M MgCl2, pH 7.0. During the elution phase two protein peaks were collected (i.e., 6 mL & 11 mL respectively) and proteins were analyzed on SDS-PAGE. The 1 peak did not contain any detectable RBD-C-tag and the 2 peak contained at least 95% pure RBD-C-tag, as determined subsequently by HPLC. The elution buffer of collected fractions containing purified RBD-C-tag was exchanged to PBS, pH 7.2 using a Spectrum MicroKros hollow fiber filter module 10 kDa (Repligen, Compton, CA, USA).

### 2.6. RBD-C-Tag Initial Purity Assessment by Reverse Phase HPLC

A Jasco^®^ HPLC system equipped with an Agilent^®^ Symmetry C4 Column (450 Å, 3.5 µm, 4.6 mm × 150 mm) was used to determine the purities of the RBD-C-tag preparations by reverse-phase chromatography in buffer A (0.1% TFA in HPLC grade water) and buffer B (0.1% TFA in acetonitrile). The column was first equilibrated with 30% buffer B at a flow rate of 1.0 mL/min and a temperature of 45 °C. After the column was fully equilibrated, 60 µL of purified RBD-C-tag, in duplicate, was injected into the system and eluted with 30% buffer B for 2 min, followed by a gradient of 30%-90% buffer B delivered over 9 min. After the gradient elution, the column was washed with 90% buffer B for 0.5 min to elute the remaining proteins. Finally, the column was re-equilibrated with 30% buffer B for 3.5 min. The HPLC run monitored proteins using absorbance at 280 nm. 

### 2.7. Glycoprotein Analyses and RBD-C-Tag Antigen Characterizations by SDS-PAGE and Anti-RBD IgG Western Immunoblotting

To evaluate the secretion of C1-RBD-C-Tag during the fermentation stage, the degree of purification during downstream processes, as well as the integrities of samples of supernatants or purified RBD-C-tag had been heated to 95 °C for 5 min in a sample buffer containing DTT (Bio-Rad, Hercules, CA, USA) and loaded onto 10-wells Precast 4–12% Bis-Tris polyacrylamide gels (Thermo Fisher Scientific, USA) together with pre-stained protein markers (Bio-Helix, Taipei, Taiwan). SDS-PGE was performed in Xcell SureLock Electrophoresis Cells (Novex, Thermo Fisher Scientific, Waltham, MA, USA) powered by PowerPac Basic (Bio-Rad, Laboratories, Hercules, CA, USA) using MOPS-SDS running buffer (Novex, Thermo Fisher Scientific, Carlsbad, CA, USA). Electrophoresis conditions were 150 Volts, 35–40 mA for 45–50 min. By the end of the electrophoresis run, these gels were either stained with InstantBlue Coomassie Protein Stain (Abcam, Fremont, CA, USA) or analyzed by western immunoblot analyses. 

For western blot analysis, iblot-2 nitrocellulose stacks (Invitrogen by Thermo Fisher Scientific, Carlsbad, CA, USA) had been used in a semi-dry I BLOT 2 Transfer Device (Invitrogen by Thermo Fisher Scientific Carlsbad, CA, USA). The blot membranes were blocked with TBST (0.05% Tween-20, 0.15 M NaCl, 0.01 M Tris-HCl buffer pH 8) containing 5% dried non-fat milk powder (Bio-Rad, CA, USA) for 1 h at room temperature. As a recognition antibody, we used rabbit anti-RBD-IgG (IIBR) generated against RBD produced in mammalian cells, purified IgG diluted 1:1000 in TBST buffer with 2.5% non-fat milk powder protein and as a reporting antibody, goat anti-rabbit-IgG (GARIG) linked to alkaline phosphatase (Jackson Immuno Research laboratories Inc., West grove, PA, USA). Membrane bound protein development was carried out by using BCIP^R^NBT-Purple liquid Substrate System for Membranes (Sigma-Aldrich, Burlington, MA, USA). 

### 2.8. RBD-C-Tag Sample Preparation, LC-MS and Data Processing

Antigen vaccine samples were subjected to in-solution tryptic digestion in the presence of urea, followed by a desalting step. The resulting peptides were analyzed using nanoflow liquid chromatography (nanoAcquity) coupled to high-resolution, high mass accuracy mass spectrometry (Q Enactive HF). Each sample was analyzed on the instrument separately in a random order in discovery mode. Raw data were processed with MaxQuant v1.6.6.0. The data were searched with the Andromeda search engine against the *Myceliophthora thermophila* proteome database, UniPort Proteome ID UP00007322, appended with our RBD-C-tag sequence and commonly recognized lab protein contaminants. The following modifications were used: carbamido-methylation of C as a fixed modification and oxidation of M and protein N-terminal acetylation as variable ones. The LFQ (Label-Free Quantification) intensities were calculated and used for further calculations using Perseus v1.6.2.3. Decoy hits were filtered out, as well as proteins that were identified on the basis of modified peptides only. The common contaminates were labeled with a ‘+’ sign in column L. The LFQ intensities were used to calculate the fold-change of the proteins between the two samples.

In total, we identified and quantified 243 proteins in both samples of RBD-C-tag. These RBD-C-tag protein preparations identified 26 peptides, achieving a sequence coverage of almost 97%. The main resultant files contained qualitative and quantitative data for each identified protein, along with the relevant relative LFQ intensity comparisons between samples. Label-free proteomics allows one to perform relative quantification between samples, but not for comparing different proteins within the same sample. Since we were primarily interested in estimating the purity of our samples, we calculated the iBAQ intensities. These values are an estimation of the absolute protein amount but are not considered an accurate quantitative measure. 

### 2.9. SARS-CoV-2 Virion Specific Antibody Monitored by Plaque Reduction Neutralization Test (PRNT)

Virus-specific neutralizing antibodies were determined by the plaque reduction neutralization test (PRNT), which is generally considered to be one of the gold standards for assessing humoral antibody-mediated correlates of immune protection for many infectious viral diseases. We monitored the ability of RBD-C-tag vaccinated BALB/c and K18-hACE2 transgenic C57BL/6J mice sera to neutralize the infectivity of SARS-CoV-2 live virus or the infectivity of rVSV-SARS-CoV-2-spike-pseudovirus (GISAID accession EPI_ISL_406862 provided by Bundeswehr Institute of Microbiology, Munich, Germany, ref; and rVSV-SARS-CoV-2-S, IIBR respectively). Blood samples from C1-RBD-C-Tag immunized or control vaccinated BALB/c and K18-hACE2 C57BL/6J mice were collected and treated as described above. Primate Vero E6 cells were seeded in 12-well plates (5 × 10^5^ cells/well) for the SARS-CoV-2 PRNT and in 6-well plates (7 × 10^5^ cells/well) for the rVSV-SARS-CoV-2-S PRNT assays and cells were grown overnight in Dulbecco’s modified eagle’s medium (DMEM) containing 10% fetal bovine serum (FBS), MEM non-essential amino acids (NEAA), 2 mM L-glutamine, 100 Units/mL penicillin, 0.1 mg/mL streptomycin, 12.5 Units/mL nystatin (P/S/N, all obtained from Biological Industries, Israel). All mouse sera (RBD-C-tag and control vaccinated mice) were heat-inactivated at 56 °C or for 30 min, then diluted in two-fold serial dilutions (between 1:20 and 1:512,000) in 400 µL of cells growth media, mixed with 400µL containing 300 PFU/mL (live virus plaque forming units) of either live SARS-CoV-2 (for BALB/c RBD-C-tag vaccinated mice) or rVSV-SARS-CoV-2-S (for hACE2 RBD-C-tag vaccinated mice), and incubated at 37 °C, 5% CO_2_ for 1 h. Cell monolayers were washed once with DMEM w/o FBS (for SARS-CoV-2 neutralization only) and 200µL of each serum–virus mixture was added in triplicate to the cells for 1 h at 37 °C. Virus mixtures without serum served as negative controls. Two milliliters per well of overlay media were added to each well and plates were incubated at 37 °C, 5% CO_2_ for 48 h. (For SARS-CoV-2) or 72 h (for rVSV-SARS-CoV-2-S). Following incubation, the media overlay was aspirated, and the cells were fixed and stained with 1 mL/well of crystal violet dye solution. The number of viral plaques in each well was determined, and the serum dilution that neutralized 50% of the infectious virions (neutralization titer, NT_50_) was calculated using Prism software (GraphPad Software Inc.).

### 2.10. RBD-C-Tag Antigen Vaccine Candidate Preparation and RBD-C-Tag Alhydrogel^®^‘85’ Gel Binding

Aluminum-based vaccine adjuvants have been successfully used in FDA-approved vaccines for the prevention of infectious diseases for over eighty years [56,57,58,59]. To prepare the RBD-C-tag antigen vaccine candidate, 3.5 mL PBS containing 290 μg/mL RBD-C-tag was added to 6.5 mL of buffered saline containing 1.6, 2.5, or 3.2 mL of 2% Alhydrogel^®^‘85’ (Brenntag Biosector A/S Denmark, acquired by Croda International PLC, UK) [60]. The aluminum (Al) content of Alhydrogel‘85’ is 10 mg/mL. Since the Alhydrogel^®^‘85’ contains 10 mg/mL aluminum oxide hydroxide, the formulated RBD-C-tag formulations contained (*v*/*v*) 16%, 25% and 32% Al, respectively. Thus, a 200 μL dose given to mice would contain 20.3 μg of RBD-C-tag and 320 μg (16%), 500 μg (25%), 640 μg (32%) of Aluminum, respectively. The RBD-C-tag Alhydrogel^®^‘85’ slurry was gently mixed for 12 h at 4 °C to ensure binding of RBD-C-tag to Alhydrogel^®^‘85’ reached equilibrium. For binding analysis, the slurry was centrifuged at 8000× *g* for 3 min and the supernatant collected. The RBD-C-tag protein content in the supernatant fraction was measured using a micro-Bradford protein assay (Bio-Rad Laboratories, Hercules, CA, USA). Thirty microliters of the supernatant samples were heated at 95 °C for 5 min in sample buffer with DTT (Bio-Rad Laboratories, Hercules, CA, USA) and proteins analyzed by SDS-PAGE and western blots, as described above. 

### 2.11. RBD Immunogenicity Assessments in Experimental Mice

Female BALB/c mice (6−8 weeks old) were obtained from Charles River Labs and randomly assigned into cages in groups of 10 animals each. Female homozygous K18-human angiotensin-converting enzyme2 (K18-hACE2 transgenic C57Bl/6 mice) were obtained from (Jackson Laboratory, Bar harbor, ME, USA) and randomly assigned into cages in groups of 8 and 6 mice. The mice were allowed free access to water and standard rodent diet (Harlan, Rehovot, Israel). Animal experiments were conducted in accordance with regulations outlined in the U.S. Department of Agriculture (USDA) Animal Welfare Act and the conditions specified in the Guide for Care and Use of Laboratory Animals, NIH. All experiments were carried out in accordance with the ARRIVE guidelines (protocol M-37-20).

### 2.12. Alhydrogel^®^‘85’, Aluminum Hydroxide Oxide Gel [AlO(OH)], Adjuvant Dose Ranging

Three groups of female BALB/c mice (10 mice per group) were immunized by subcutaneous (SC) needle injection with Alhydrogel**^®^**‘85’ formulated RBD-C-tag antigen at day 0 (20 μg RBD-C-tag), day 20 (20 μg RBD-C-tag), and day 41 (20 μg RBD-C-tag) (Figure 2a). In group 1, the 0.2 mL volume of each inoculation formulation of Alhydrogel**^®^**‘85’ [AlO(OH)] contained a total of 320 μg Alhydrogel**^®^**‘85’ and 20 μg RBD-C-tag; in group 2, the formulation contained a total of 500 μg Alhydrogel**^®^**‘85’ and 20 μg RBD-C-tag; and in group 3, the formulation contained a total of 640 μg Alhydrogel^®^‘85’ and 20 μg RBD-C-tag was used. Alhydrogel**^®^**‘85’ alone was used as a negative control. At days 19, 27, 40 and 51 blood was collected and sera were evaluated for anti-RBD-C-tag immunoglobulin and virus neutralizing antibody titers. 

### 2.13. K18-hACE2 Transgenic C57BL/6J Mouse Challenge Tests with Lethal Live SARS-CoV-2 Virus

All animal experiments involving SARS-CoV-2 were conducted in a BSL3 facility. Infection experiments were carried out using SARS-CoV-2, isolate Human 2019-nCoV ex China strain BavPat1/2020 kindly provided by Dr. Christian Drosten (Charité, Berlin, Germany) through the European Virus Archive—Global (EVAg Ref-SKU: 026V-03883). The original viral isolate was amplified by five passages and quantified by plaque titration assay in Vero E6 cells and stored at −80 °C until use. The viral stock RNA sequence and coding capacity were confirmed. SARS-CoV-2 BavPat1/2020 virus diluted in PBS supplemented with 2% FBS (Biological Industries, Israel) was used to infect animals by intranasal instillation of mice anesthetized by intraperitoneal injection of ketamine (160 mg/kg) and xylazine (6 mg/kg). Female homozygous K18-human Angiotensin-Converting Enzyme2 (K18-hACE2 transgenic C57BL/6J mice) were obtained from Jackson Laboratory (ME, USA) and randomly assigned into cages in groups of eight mice. The mice were allowed free access to water and standard rodent diet (Harlan, Rehovot, Israel). Virus infections were performed according to the standard operating procedures of the BioSafety Level-3 (BSL-3) facility.

Treatment of animals was in accordance with regulations outlined in the U.S. Department of Agriculture (USDA) Animal Welfare Act and the conditions specified in the Guide for Care and Use of Laboratory Animals, NIH. All experiments were carried out in accordance with the ARRIVE guidelines. Animal studies were approved by the IIBR animal care and use ethical committee (IACUC) on animal experiments (protocol M-64-20). 

K18-hACE2 transgenic C57BL/6J mice are an excellent model for SARS-CoV-2 pathogenicity and to measure protection efficacy of vaccine and therapeutic antiviral agents against morbidity, mortality, cytokine and chemokine storms, histopathology, and viral replication [61,62,63]. Transgenic mice expressing human angiotensin-converting enzyme 2 (hACE2) under control of the human cytoKeratin18 promoter (K18-hACE2) succumbed to SARS-CoV-2 infection by day six, with virus in the lung airway epithelium and brain. hACE2 expression in K18-hACE2 C57BL/6J mice occurred in airway epithelial cells, where SARS-CoV-2 infections are typically initiated. K18-hACE2 C57BL/6J mice produced a modest Th1/2/17 cytokine storm in lung and spleen that peaked by day two, with extended chemokine storms in both lungs and brain, which was detected in the brain at day six [63]. 

To determine whether a single or a double boost of 20 μg C1-RBD-C-tag/mouse is sufficient for protection of female K18-hACE-2 C57BL/6J mice against a challenge with 2000 PFU of live SARS-CoV-2 (Figure 2b,c), two groups of eight K18-hACE-2 C57BL/6J mice were vaccinated SC with 20 μg C1-RBD-C-tag/mouse formulated with 640 μg Alhydrogel^®^‘85’and then single-boosted (Figure 2b) or double-boosted (Figure 2c) with 20 μg C1-RBD-C-tag/mouse formulated with 640 μg Alhydrogel^®^‘85’ at 21 (20 μg RBD) or at days 21 (20 μg RBD) and 42 (20 μg RBD) post-prime vaccination, respectively. 

As placebo controls, two groups of 3 K18-hACE-2 C57BL/6J mice were vaccinated in the same manner with 640 μg Alhydrogel^®^‘85’. At 21 days post the 1 boost (Figure 2b) and 14 days post the 2 boost (Figure 2c), the RBD-C-tag and placebo vaccinated mice were anesthetized and challenged intranasally (IN) with 2000 PFU live lethal SARS-CoV-2 per mouse, which corresponds to 200× the SARS-CoV-2 lethal dose (LD_50_) (SARS-CoV-2 GISAID accession EPI_ISL_406862 provided by Bundeswehr, Institute of Microbiology, Munich, Germany). 

General observations for morbidity and weight loss of challenged animals were carried out for three weeks post-challenge, and the final blood serum was harvested from all surviving mice. At days 20, 35, 61 (Figure 2b), and days 20, 41, 54, 77 (Figure 2c) mouse peripheral blood was harvested, and sera were evaluated by ELISA for anti-RBD-C-tag antibody titers, and virus plaque reduction neutralization titer (PRNT) was measured to determine neutralizing antibody titers.

### 2.14. Antibody Iso-Type Immunoglobulin G (IgG) Class Subtyping of DYAI-100A85 Vaccinated K18-hACE2 Transgenic C57BL/6J Mice

Analysis of specific anti-RBD-C-tag IgG subtypes was carried out using a standardized mouse immunoglobulin isotyping panel, SAB Clonotyping System C57BL/-HRP (Horse Radish Peroxidase; Southern Biotech, Birmingham, AL, USA) after each antigen boost and post-challenge of the K18-hACE2 vaccinated C57BL/6J mice. Blood samples were collected from inoculated mice and treated as described above. Recombinant RBD and purified mouse IgG1, IgG2b, IgG2c and IgG3 (Southern Biotech, Birmingham, AL, USA) served as positive controls and was used to coat ELISA plates. Serially diluted mouse sera were added and incubated at 37 °C for 1 h, and then the plates were washed three times. HRP-conjugated anti-mouse IgG, IgG1, IgG2b, IgG2c and IgG3 was diluted 1:1000 in blocking solution and added to wells (50 μL/well). After incubation for 1 h at 37 °C, plates were washed three times and developed with 3,3′,5,5′-TetraMethyl Biphenyl diamine (TMB, Sigma-Aldrich, Burlington MA, USA) for 20 min. The reactions were stopped with 50 μL/well of 1.0 M H_2_SO_4_ and the optical density at 450 nm was determined using a SpectraMax iD3 (Molecular Devices, San Jose, CA, USA) and SoftMax v7.0 software. 

### 2.15. Statistical Analysis

Statistical analyses were performed using Prism 9.3.1 (GraphPad software) and Stats tester mini. Comparisons among multiple groups were performed using one-way ANOVA with Tukey’s multiple comparison pos hoc test. Comparisons between two groups were performed using unpaired Student’s *t*-tests. *p* values of <0.05 were considered significant. 

## 3. Results

### 3.1. Production of the SARS-CoV-2-Spike-S1^333–527^RBD-C-Tag Glycoprotein Recombinant Subunit Antigen in Thermothelomyces Heterothallica

First, the original D614G (Wuhan-Hu-1) isolate of SARS-CoV-2-spike-S1***^333–^****^527^*RBD (receptor binding domain) glycoprotein vaccine antigen candidate was selected, as described in Methods. A similar S1^332–532^RBD monomeric, glycan-engineered fragment was previously shown to be more highly expressed and thermo-tolerant and was expressed at a purified yield of 214 mg/Liter in an optimized, mammalian cell culture [43]. In contrast to the stabilized spike ectodomain, this S1^332–532^RBD region is stable when heated at temperatures up to 100 °C, when lyophilized; or when heated up to 70 °C, when in-solution; and is also stable for over 4 weeks at 37 °C, as monitored by surface plasmon resonance, SPR-sensograms of ACE2 binding and by RBD protein integrity, as evidenced in sodium dodecyl sulphate-polyacrylamide gel electrophoresis analyses (SDS-PAGE).

The SARS-CoV-2-S1**^333–^**^527^RBD-C-tag vaccine candidate is a 201 amino acid long polypeptide, encoding the SARS-CoV-2-RBD amino acid residues 333–527, with an additional four amino acid C-tag and a two-amino acid linker at the C-terminus. The DNA encoding the S1**^333–^**^527^RBD-C-tag was synthesized and cloned into an expression vector under the native *bgl8* promoter. The expression vector was integrated by homologous DNA recombination into the *bgl8* locus yielding high expression, in the strain DNL155 where 14 protease genes have been deleted. The transformed C1 cells were screened for production of the vaccine candidate with 24-well plate cultures and western blotting. Positive transformants were purified through a single colony culture and shown by PCR to have correct integration of the expression construct and no remaining parental strain genetic material. 

The selected C1 producing strain was grown in a standard 10 L fermentation as described in the Methods section. During development, fermentation yields were continually evaluated by analyzing secreted proteins by SDS-PAGE. The expected approximately 24 kDa SARS-CoV-2 spike S1**^333–^**^527^RBD-C-tag protein was detected, at 29 h. of elapsed fermentation time (EFT) and later time points (Figure 3, lanes 2–6). RBD-C-tag protein was harvested at, or near, peak production at 90 h. EFT (Figure 3, lane 6), where production level was approximately 1 g/L. Glucose medium feeding was started at 36 h. EFT (Figure 3, lane 3) and the media was adjusted to maintain a packed cell volume (PCV) ratio of 25% (packed cell *weight*/*volume*).

### 3.2. Integrity and Purity of RBD-C-Tag Subunit Recombinant Antigen Vaccine Harvested at 90-h EFT, as Monitored by Protein SDS-PAGE

The C1-produced S1**^333–^**^527^RBD-C-tag vaccine candidate was purified with C-tag affinity chromatography as described in Methods. The affinity purified protein was analyzed for purity by SDS-PAGE (Figure 4). Affinity chromatography purified RBD-C-tag was loaded onto an SDS-PAGE gel and proteins stained with Coomassie blue dye showed a single homogeneous protein product at about 24 kDa. under reducing conditions. 

### 3.3. LC-MS-MS Analysis of the Purity of the Affinity Purified S1^333–527^RBD-C-Tag Glycoprotein Subunit Antigen

LC-MS/MS analyses of the purified RBD samples provided another measure of vaccine antigen purity. The qualitative and quantitative data for each identified protein, along with the relevant relative label-free quantification (LFQ) intensity comparisons between samples is based on the calculated, intensity-based absolute quantification (iBAQ). iBAQ is defined as the total intensities divided by the identified peptides for one protein. It is difficult to perform an absolute quantification of proteins, however, iBAQ assumes that all oligopeptides are ionized and detected at about the same efficiencies. LFQ is very similar to the “protein intensities” used by iBAQ (i.e., the sum of peak intensities of all peptides of a protein/protein group divided by the number of theoretically observable peptides) and can provide the absolute protein amounts. The purity of the RBD-C-tag vaccine antigen following C-tag affinity chromatography protein purification, was determined as approximately 97% pure. The RBD-C-tag protein was identified as containing 28 oligopeptides, achieving a sequence coverage of almost 95%. The consistent results generated from these purified samples, taken together, indicated that the RBD-C-tag vaccine antigen is of high purity.

### 3.4. HPLC-Reverse Phase Chromatography Analysis of Purity of the Affinity Purified S1^333–527^RBD-C-Tag Glycoprotein Subunit Antigen

The purity of the final RBD-C-tag antigen vaccine product candidate was further analyzed by HPLC reverse phase affinity chromatography. The chromatogram is shown in (Figure 5). Based on these and other similar repeat results, the purity for these two runs of purified RBD-C-tag antigen was calculated as achieving approximately 97.4% purity consistent with the purity determined by SDS-PAGE with Coomassie blue dye staining (Figure 3).

### 3.5. RBD-C-Tag Antigen Binding to Recombinant Human ACE2 (Angiotensin Converting Enzyme 2)

Using ELISA to monitor the protein-binding activity between the RBD-C-tag recombinant antigen and recombinant human ACE2 (hACE2), we determined that the RBD-C-tag antigen produced in C1-cells specifically bound to the hACE2 receptor in a dose-dependent manner and with the same specificity and efficiency, as directly compared to the reference preparation of RBD antigen produced in CHO cells (data not shown). This and other similar repeat experiments indicated that the C1-produced RBD-C-tag recombinant subunit antigen specifically and strongly binds hACE2 receptors and, thus possesses RBD specific antigen epitope conformation(s) functional for hACE2 receptor binding. 

### 3.6. S1^333–527^RBD-C-Tag Recombinant Subunit Antigen Binding to Aluminum Oxide Hydroxide, Alhydrogel^®^‘85’ Adjuvant

The binding of the RBD-C-tag antigen to the Alhydrogel^®^‘85’ adjuvant was monitored to measure the extent of binding of aluminum oxide hydroxide gel to the RBD-C-tag antigen. The protein content in the supernatant fraction after adsorption and centrifugation was measured in the RBD subunit antigen vaccine preparations at adjuvant:antigen (*weight*:*weight*) ratios of 16:1, 25:1, and 32:1. These measurements showed that no detectable protein antigen remained in the supernatant fraction after Alhydrogel^®^‘85’ adjuvant adsorption, indicating that in all three tested ratios, all (100%) of the RBD-C-tag vaccine antigen was bound to the Alhydrogel^®^‘85’ adjuvant Appendix A. 

### 3.7. Alhydrogel^®^‘85’ Adjuvant Dose Range Screening of Adjuvanted-RBD Subunit Antigen Vaccine Immunogenicity in Mice

Alhydrogel^®^‘85’ was carefully selected to serve as the preferred vaccine adjuvant on the basis of its well-documented record of effectiveness and safety [56,57,58,59]. It is readily available in large bulk GMP quantities at a very low-cost for worldwide approved uses in both human and animal vaccines and it is known to be compatible with other innate and adaptive immune modulating additional adjuvants. Alhydrogel^®^‘85’ as a vaccine adjuvant was also specifically selected for lowering the amount of the subunit glycoprotein antigen dose (dose-sparing) requirement for DYAI-100A85, the RBD-C-tag vaccine candidate. 

Combining Alhydrogel^®^‘85’ as the aluminum oxide hydroxide gel adjuvant with recombinant RBD-C-tag antigen was fully expected to reduce the amount of RBD-C-tag antigen required to induce protective antibody and anti-viral host immune responses, thus reducing the dose administered of the adjuvanted-vaccine. In addition, Alhydrogel^®^‘85’, together with an additional adjuvant such as ASO3 or CpG 1018, may also enable a balanced Th1/Th2 immunity [41,43].

We performed careful comparisons of the different ratios of Alhydrogel^®^‘85’ to RBD-C-tag in BALB/c mice immunized with three subcutaneous injections of Alhydrogel^®^‘85’ formulated with RBD-C-tag antigen given on days 0, 21, and 41. We systematically compared a 20 μg/dose of RBD-C-tag antigen adjuvanted with either 320 μg Alhydrogel^®^‘85’, or 500 μg Alhydrogel^®^‘85’, or 640 μg Alhydrogel^®^‘85’; corresponding to weight-to-weight (wt:wt) ratios of Aluminum:RBD-C-tag of 16:1, 25:1, and 32:1, respectively. Peripheral blood sera from mice were collected at days 19, 27, 40 and 51 to evaluate efficacy, as monitored by the percentage of mice that developed measurable serum antibody responses against C1-RBD-C-tag antigen (Figure 6), ELISA for RBD-C-tag-specific binding antibodies (Figure 7) and by PRNT (plaque reduction neutralization titer) assays for neutralizing antibodies against SARS-CoV-2. Sera obtained on day-19 after the first injection of 320 μg Alhydrogel®‘85’ adjuvanted RBD-C-tag (16:1 Vaccine) showed no response and did not differ from the sera obtained from the control mice injected with placebo (control) Alhydrogel^®^‘85’ alone. 

By contrast, immune sera obtained from mice that were injected with 500 μg (25:1 Vaccine) or 620 μg (32:1 Vaccine) Alhydrogel^®^‘85’ adjuvanted RBD-C-tag showed elevated IgG responses, as measured by the anti-RBD ELISA (Figure 7). *p* values were determined by two-way Analysis of Variance (ANOVA). The *p* values of placebo or 16:1 vaccine groups versus 25:1 vaccine or 32:1 vaccine groups were *p* < 0.013, while the *p* value between 25:1 vaccine and 32:1 vaccine was *p* = 0.49. DYAI-100A85 vaccine-immunized mice sera obtained on day-27 and on day-40, (corresponding to six and twenty days after the second dose of the 1:16 vaccine, respectively) consistently showed high RBD specific IgG titers. The *p* value of placebo versus 16:1 vaccine group was *p* < 0.013. Comparison of the 16:1 vaccine group to 25:1 and 32:1 vaccine groups gave *p* values of *p* = 0.336 and *p* = 0.242 for day-27 sera and *p* = 0.69 and *p* = 0.0061 for day-40 sera, respectively (Figure 7). The sera obtained at day-51, 10 days after the third vaccination gave very high titers of specific anti- RBD IgG, with Geo-Mean titers of 5572; 36,204; and 54,874 for the 16:1, 25:1 and 32:1 vaccine groups, respectively. At that stage, there was no significant difference between 25:1 and 32:1 formulated vaccines with a *p* value of 0.18, but there was a marked difference between 25:1 and 32:1 formulated vaccines to the 16:1 vaccine group with *p* values of 0.008 and 0.0071, respectively (Figure 7). 

The immune sera from day-51 of 25:1 and 32:1 formulated vaccine groups were tested for neutralizing activity against SARS-CoV-2, and a very high level of activity was consistently observed. The determined geo-mean neutralization activity with a 50% neutralization titer (NT_50_) was observed at reciprocal dilutions of 14,929 for the 25:1 vaccine and of 10,196 for the 32:1 vaccine. In summary, these, and other similar repeat measurements showed that BALB/c mice immunized with RBD-C-tag antigen formulated as Alhydrogel^®^‘85’ adjuvanted RBD-C-tag (DYAI-100A85), in *weight*:*weight* ratios of 25:1 and 32:1, consistently produced higher neutralizing antibody titers and higher RBD specific IgG titers than those mice in the group vaccinated with the formulation of RBD:Alhydrogel^®^‘85’ ratio of 16:1. 

### 3.8. Subcutaneous Prime and Boost Vaccine Inoculations of Alhydrogel^®^‘85’ Adjuvanted RBD-C-Tag Antigen (DYAI-100A85) Protect k18-hACE2 Transgenic Mice against Intra-Nasal Challenge with Lethal SARS-CoV-2 Live Virus

To determine the immune-protective efficacies of Alhydrogel^®^‘85’ adjuvanted RBD-C-tag vaccine, we used K18-hACE2 C57BL/6J transgenic mice as the animal model of disease and mortality for SARS-CoV-2-induced Acute Respiratory Distress Syndrome (ARDS) [61,62,63]. Two groups of K18-hACE2 C57BL/6J mice were immunized. The first experimental group of mice was immunized subcutaneously on days 0 and 21 with a total of two injections, each injection containing 20 μg RBD-C-tag adjuvanted with 640 μg Alhydrogel^®^‘85’) adjuvant:antigen ratio of 32:1). These mice were challenged on day 41, 20-days after the first Boost (2nd injection total), by intranasal inoculation of mice with 2000 PFU (Plaque Forming Units) of lethal live SARS-CoV-2 virus. The second experimental group of mice was immunized subcutaneously on days 0, 21, 42 with a total of three injections, each injection containing 20 μg RBD-C-tag adjuvanted with 640 μg Alhydrogel‘85’^®^. These mice were challenged 14 days after the third injection (corresponding to the 2nd booster), by intranasal challenge with 2,000 PFU of lethal live SARS-CoV-2 virus. The control mice groups were injected with 640 μg of Alhydrogel^®^‘85’ adjuvant alone as a “negative control” without vaccine RBD antigen. Peripheral blood sera were collected on days 20 and 35, for the first (A), group and on days 20, 41 and 54, for the second (B), group to evaluate efficacy, as monitored by ELISA for RBD-C-tag-specific binding antibodies (Figure 8), and by PRNT (plaque reduction neutralization titer) assays for neutralizing antibodies against VSV-SARS-CoV-2-S1 pseudo-type virus (Figure 9). In sera obtained from mice of the 1st group, 14-days post-second immunization, the reciprocal titers of SARS-CoV-2 RBD-specific IgG ranged from 3200 to 102,400 with a geometric mean titer (GMT, Geo-Mean, reciprocal) of 16,600. (Figure 8A) *p* values were determined by two-way Analysis of Variance (ANOVA). The *p* value of reciprocal-titers determined for the sera obtained on day 20 versus titers of 14 days post boost was *p* = 0.013. The determined PRNT_50_ values ranged to 32,000 with a GMT of 5233 (Figure 9). The prime-boost immunization regime protects 87.5% of mice, due to a single mouse that did not develop a sufficient immune response (Figure 10). In sera obtained from mice of the second group, 20-days post-second immunization, the reciprocal titers of SARS-CoV-2 RBD-specific IgG ranged from 3200 to 102,400 and the sera obtained at day 54, 14 days after the third vaccination elicited very high titers of specific anti-RBD IgG, that ranged from 25,600 to 409,600 with geo-mean titers of 157,922 (Figure 8B). The third immunization elicited very high levels of antiviral neutralizing antibodies in all mice, with PRNT_50_ values in the range of 22,627–512,000 with a GMT of 76,108 (Figure 9). The *p* value of reciprocal-titers determined for the sera obtained on day 41, versus titers at 14-days post the 2nd boost was determined as *p* = 0.0014. The prime boost, boost immunization regime protected 100% of mice from the challenge with lethal live virus (Figure 10). Moreover, there was no difference in the mean body weight of mice observed in all study groups, thus indicating that the C1-produced RBD-C-tag antigen-treated and control adjuvant only-treated mice did not develop any adverse effects toxic effects at the doses investigated.

### 3.9. Serum IgG Subtyping of DYAI-100A85 Vaccinated K18-huACE2 Transgenic C57BL/6J Mice

Antigen-specific serum IgG1 and IgG2b antibodies were measured in sera obtained from K18-huACE2 transgenic C57BL/6J mice of the first group, 14 days post-second immunization. The reciprocal-titers of SARS-CoV-2 RBD-specific IgG1 were in the range of 1280 and IgG2b in the range of 40 or less. In sera obtained 20-days after the Intra-Nasal challenge with live SARS-CoV-2, titers of IgG1 ranged from 8000 to 128,000 and IgG2b ranged from 100–800. In the sera obtained from mice of the second group 14 days after the third vaccination, the reciprocal-titers of SARS-CoV-2 RBD-specific IgG1 were in the range of 12,800 and IgG2b was in the range of 40 to 160. In sera obtained 21 days after the intranasal challenge with SARS-CoV-2, titers of IgG1 ranged from 128,000 to 256,000 and titers of IgG2b ranged from 200–800. IgG1 is one of the most abundant IgG subclasses in human sera and it is important for mediating antibody responses against viral pathogens. It does so by binding to soluble proteins and membrane protein antigens via its variable domain and concomitantly activating effector mechanisms of the innate immune system. IgG1 is associated with Th2 response while IgG2b is associated with Th1 response [64], and thus it is evident that DYAI-100A85 elicited mainly Th2 immune response and did not promote significant Th1 cellular response. 

## 4. Discussion

The results of the studies presented here demonstrate the feasibility and practicality of rapidly and economically developing a safe, effective, protective, and inexpensive recombinant vaccine candidate based on SARS-CoV-2 receptor-binding domain (RBD) glycoprotein expressed and produced in *Thermothelomyces heterothallica* (C1 strain DNL155). C1 strain DNL155 was rapidly genetically engineered to secrete high levels of recombinant SARS-CoV-2-S1**^333–^**^527^RBD-C-tag receptor binding domain for rapid and cost-effective affinity purification. The recombinant DNA expression cassette includes a strong constitutive promotor for efficient transcription, glycoprotein synthesis, and antigen secretion into the fermentation media in a completely characterized 14-protease-deficient C1 clonal strain. Fermentation development and optimization, together with the development of an efficient and commercially scalable downstream processing using the CaptureSelect™ C-tag affinity matrix, resulted in yields of 800 mg/L of C1-RBD-C-tag with more than 97% purity in fermentation runs of four days’ duration (Figure 3, Figure 4 and Figure 5). Further fermentation optimization has resulted in improved production levels of over 2 g/L.

Characterization of the glycoprotein structure of the recombinant RBD-C-tag^333–527^ produced in C1 was previously compared to RBD produced in yeast (*P. pastoris*), bacterium (*E. coli*), and mammalian cells (CHO and HEK 293T), using buffer-free protein digestion with electro spray ionization-mass spectrometry (ESI-MS) analyses [65]. The characterization confirmed its identity with the typical heterogeneity of N-glycans and the expected molecular masses, as well as the integrity of the N- and C-terminal ends and formation of the four disulfide bonds. It is important to note that stable hACE2 binding activities were obtained in vitro and in vivo for this thermophilic filamentous fungus-C1-cell-clone produced SARS-CoV-2-spike-S1*^333–527^*RBD-C-tag-Glycoprotein subunit antigen after three days incubation at room temperature (25 °C).

The host immune responses elicited by the purified S1^333–527^RBD-C-tag vaccine antigen produced in C1-cells were evaluated both in BALB/c and K18-hACE2 transgenic C57BL/6J mice after optimization of the adjuvanted antigen vaccine formulation with Alhydrogel^®^‘85’ (Figure 7 and Figure 8). We have demonstrated that DYAI-100A85 vaccine candidate formulated as an Alhydrogel^®^‘85’ adjuvanted SARS-CoV-2 spike S1^333–527^RBD-C-tag antigen elicits strong neutralizing anti-SARS-CoV-2 IgG antibodies with protective immunity against challenge with SARS-CoV-2 live virus infection, as monitored in K18-hACE-2 transgenic C57BL/6J mice (Figure 9 and Figure 10). 

In further pre-clinical studies under GLP conditions [54], we consistently observed no local reactogenicities and no systemic toxicities of DYAI-100. In summary, no signs of adverse events and toxicity were observed in all experiments, including injection site reactions. Furthermore, New Zealand white rabbits, treated under GLP conditions with four repeated weekly injections of DYAI-100, developed follicular hyperplasia in the spleen and inguinal lymph nodes. This hyperplasia was shown to consist of B cells by immunohistochemistry and was accompanied with an increase in anti-SARS-CoV-2 specific IgG antibodies. These changes persisted throughout the study recovery period (42 days post first dosing, Figure 11), suggesting that DYAI-100 provides sustained immunogenic response against RBD-C-tag. No local or systemic toxicity was observed in the toxicological evaluation study [54]. Therefore, these and other similar repeated results indicate that DYAI-100 appears to possess an excellent safety profile, as demonstrated in vivo in experimental animals. Our vaccine candidate has comparable efficacy and safety to existing authorized and licensed recombinant RBD vaccines. Based on the data DYAI- 100 receives regulatory approval to initiate a Phase 1 clinical trial to demonstrate safety and efficacy in humans [55].

Our strategy for manufacturing recombinant subunit antigen vaccine in C1 production platform is generally applicable and adaptable to variety of different and newly evolving highly infectious and pathogenic respiratory viruses. This includes SARS-CoV-2-Omicron and other VOCs, as well as seasonal influenza or respiratory syncytial viruses and other mutant influenzas (variant influenza clades) [52,53,65,66]. The DNL155 and other advanced strains of the C1 platform can rapidly produce complex recombinant glycoprotein antigens with the appropriate post-translational carbohydrate modifications [66,67] and are cultivated significantly faster, as compared to other cell-based vaccine manufacturing systems (i.e., with a doubling time of 2.5 h for C1-cells vs. 24 h for CHO-cells). This enables production of about three or four batches of C1 produced antigen vaccine in the time it takes to propagate just one batch of CHO cell-produced antigen vaccine. Moreover, antigen production in C1 production platform uses inexpensive fermentation media that is up to 20- to 50-fold-less expensive than CHO cell culture media (relatively expensive serum-free media that contains growth factors). These advantages for C1 recombinant antigen production make C1-cells vaccine manufacturing a highly desirable, cost-effective, and ultra-rapid alternative for production of low-cost human and animal vaccine glycoprotein antigens.

## 5. Conclusions

The results clearly show that the C1 production platform is suitable for the production of glycoprotein subunit vaccines.

The C1- production platform enable a straightforward manufacturing that is both inexpensive, scalable, and easily transferable for local production, offering strong competitive advantages over most, or potentially even all other existing recombinant glycoprotein vaccine manufacturing technologies [52,53,65,66,67]. Quality-controlled QA regulatory processes are already written and validated, enabling a potentially shortened regulatory path through clinical trials and for bringing product candidates to the market. This TFF-C1-cell antigen manufacturing strategy may be generally applicable and easily adaptable for a variety of different newly evolving pathogen recombinant protein vaccines, including SARS-CoV-2-VOCs and seasonal human influenza A and B viruses [52,53,66], as well as for mutant influenzas (variant influenza clades). SARS-CoV-2-VOC RBD-C-tag antigen vaccine candidates (i.e., α-, β-, γ-, Δ-, Omicron, other VOCs) may be rapidly manufactured at relatively low costs. The DYAI-100A85 vaccine candidate characterized here, and other VOC RBD antigen candidates, could potentially be produced from C1-cell clones in large quantities (i.e., a few hundred kilos) sufficient to supply the necessary prime and boost vaccine doses to alleviate current stresses on global human vaccine manufacturing and supplies, and to resolve thermo-stability/cold-chain limitations. The DYAI-100A85 vaccine candidate can be stored and shipped at relatively higher (i.e., ambient, room or higher) temperatures, as compared to DNA-encoded or Messenger-RNA-encoded vaccines that need to be shipped and stored at minus 20–60 degrees Celsius. 

In summary, glycoprotein vaccine manufacturing C1 production platform appears entirely suitable for rapid commercial development of SARS-CoV-2 vaccine, which would be relatively unaffected by lack of resources, such as shortages of materials (e.g., plastic bags for insect cell vaccine manufacturing), equipment, expensive media, and cold-chain freezers. Based on the data of this study and further development under GMP conditions for the first time, a subunit antigen vaccine produced in C1-cells received regulatory approval to initiate a Phase 1 clinical trial to demonstrate safety and efficacy in humans [55].

## Figures and Tables

**Figure 1 vaccines-10-02119-f001:**
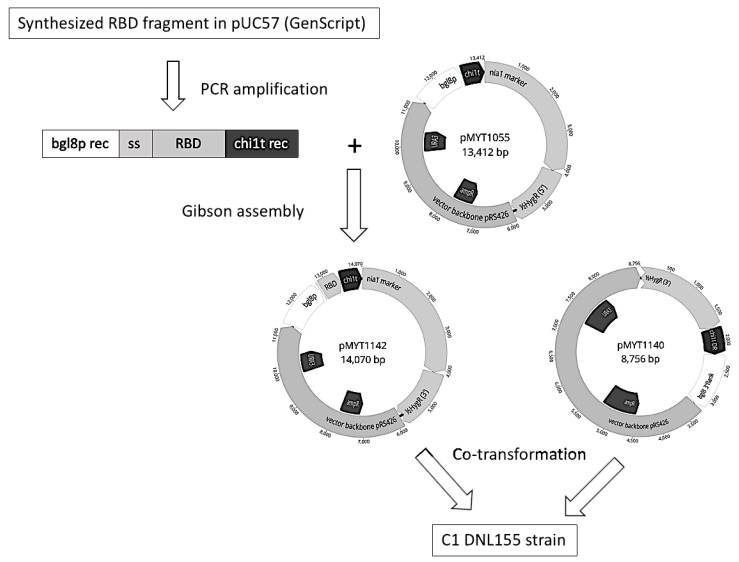
Flow chart of RBD expression plasmid pMYT1142 construction. Plasmid numbers and size as base pairs (bp) indicated. Abbreviations: bgl8p, promoter; ss, signal sequence; RBD, Receptor Binding Domain encoding amino acids 333-527 of the Spike protein from SARS-CoV-2; chi1t, terminator; bgl8p rec, 40 bp recombination sequence to bgl8p; chi1t rec, 40 bp recombination sequence to chi1t; HygR, hygromycin marker; URA3, auxotrophic marker; ampR, ampicillin marker; chi1t DR; direct repeat fragment of chi1t.

**Figure 2 vaccines-10-02119-f002:**
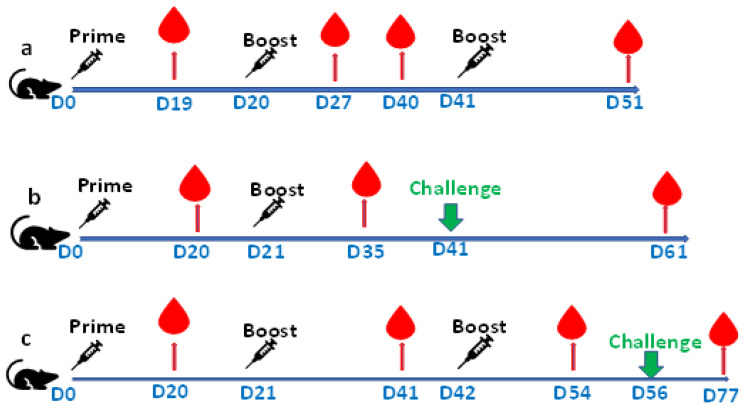
Vaccination of BALB/c (H-2^d^) female and K18-hACE2, C57Bl/6J (H-2^b^) female mice with affinity purified C1-RBD-C-tag adjuvated with Alhydrogel^®^‘85’: (**a**) three groups of BALB/c mice (10 per group) were injected subcutaneously (SC), with 20 μg of C1-RBD-C-tag/mouse adjuvanted with 320 μg, 500 μg, or 640 μg Alhydrogel^®^‘85’. As negative placebo controls, three groups of three mice were vaccinated with 320 μg, 500 μg, 640 μg Alhydrogel^®^‘85’ alone. All groups were primed and double boosted at 20 and 41 days post-prime, respectively; and (**b**,**c**) two groups of eight individual female K18-hACE2 C57BL/6J mice were primed and boosted either once (**b**) or twice (**c**) with 20 μg RBD-C-tag/mouse with 640 μg Alhydrogel^®^‘85’ at day 21 (**b**) and at days 21 and 42 (**c**). As negative controls, two groups of three individual K18-hACE2 C57BL/6J mice were each vaccinated with 640 μg Alhydrogel^®^‘85’ alone in the same manner as in (**b**,**c**). At 20 days post-injection with the 1 boost (**b**) and 14 days post the 2 boost (**c**) all C1-RBD-C-tag and control vaccinated mice were anesthetized and challenged Intra-Nasally (IN) with 2000 PFU of live SARS-CoV-2/mouse. General observations for monitoring morbidity and weight loss of the live virus challenged mice were carried out for three weeks post-challenge, and the final blood collection of all surviving mice was performed.

**Figure 3 vaccines-10-02119-f003:**
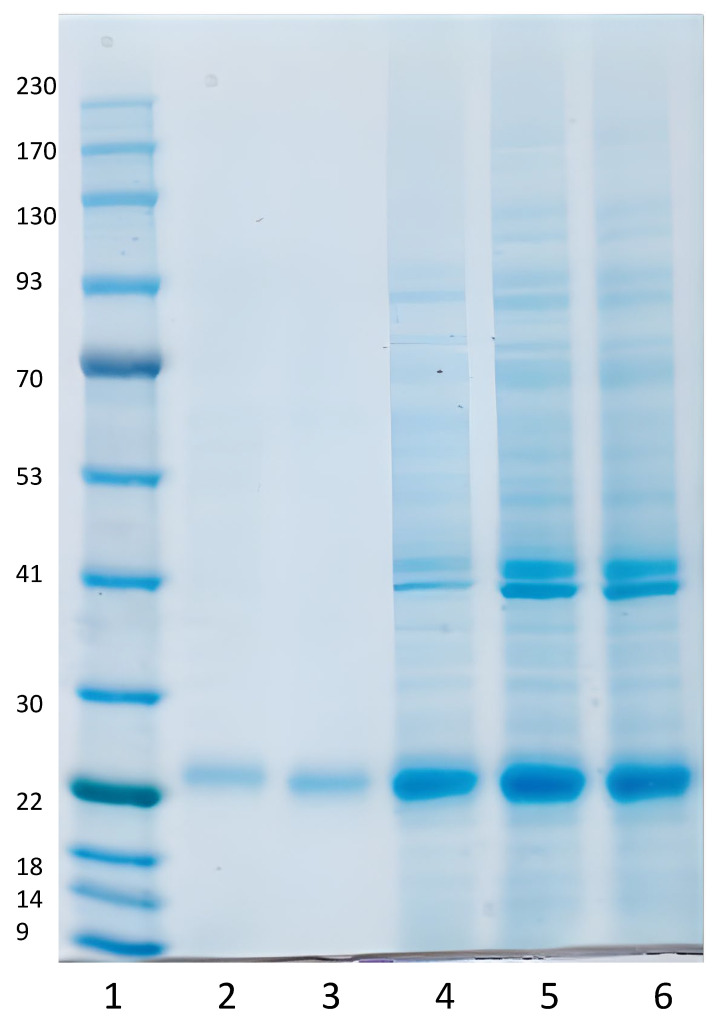
Time course of RBD*^333–527^*-C-tag vaccine subunit antigen production during the fungal C1-cell fermentation manufacturing process. Fermentation supernatant samples were analyzed in SDS-PAGE stained with Coomassie Blue. Lane 1, contains 12 different molecular weight markers, including a 22 kDa. molecular weight protein marker for reference (marked with an arrow on the left); lane 2, 29 h. of elapsed fermentation time (EFT), shows a single protein staining band at approximately 24 kDa.; lane 3, 36 h. EFT; lane 4, 60 h. EFT; lane 5, 84 h. EFT; and lane 6, 90 h. EFT.

**Figure 4 vaccines-10-02119-f004:**
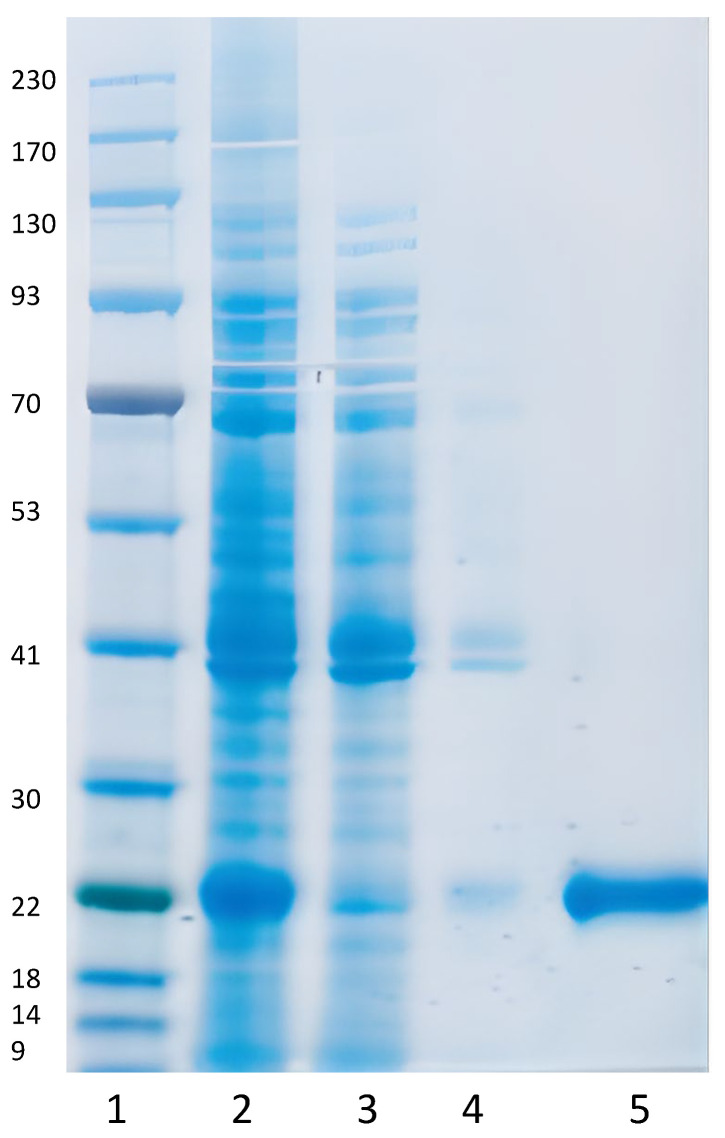
S1^333–527^RBD-C-tag vaccine antigen purification by affinity chromatography on a CaptureSelect^TM^ C-tag affinity matrix (Thermo Fisher Scientific, Carlsbad, CA, USA). The representative gel shows SDS-PAGE with Coomassie Blue dye staining of proteins. Lane 1, MW protein markers; lane 2, EFT 90 h. fermentation supernatant, lane 3, fluid flow through; lane 4, wash fraction, lane 5 2.5 µg of purified eluted RBD C-tag antigen. The purified S1^333–527^RBD-C-tag vaccine antigen is marked with an arrow on the right.

**Figure 5 vaccines-10-02119-f005:**
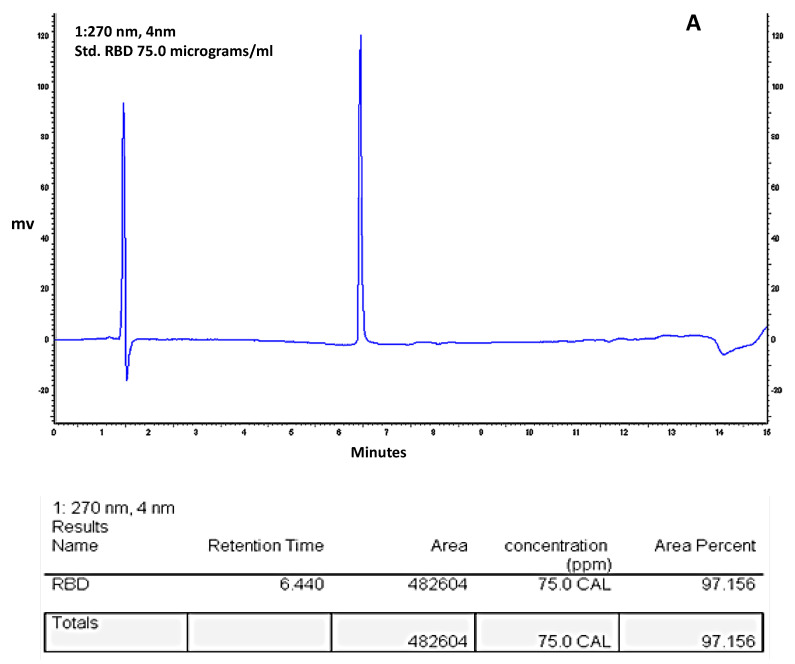
Assessment of purity of C1-cell manufactured RBD-C-tag, as determined by reverse phase HPLC-reverse phase chromatography. The purity of the two different RBD sample runs were 97.16% for the sample with 75 μg RBD (**A**) and 97.63% for the sample with 150 μg RBD (**B**). part per million (ppm), Calculated (CAL), millivolts (mv).

**Figure 6 vaccines-10-02119-f006:**
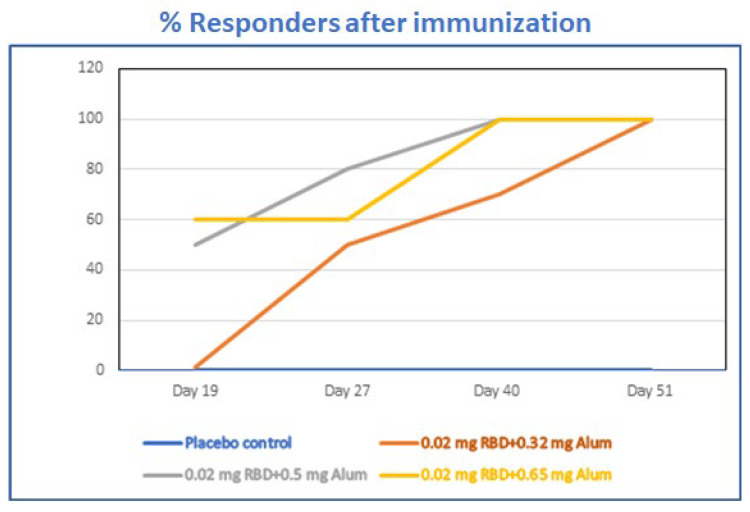
Precent BALB/c mice that developed measurable serum antibody responses against C1-RBD-C-tag antigen. The reciprocal end-point titers of anti-RBD-C-tag immunoglobulin antibody in sera from BALB/c mice immunized subcutaneously with 20 μg RBD-C-tag antigen per mouse, adjuvanted with either 320 μg, 500 μg, or 650 μg Alhydrogel^®^‘85’ representing Alhydrogel^®^‘85’/RBD-C-tag, adjuvant:antigen ratios of 16:1, 25:1 and 32:1 (*w*/*w*), respectively were measured. Titer values determined for peripheral blood sera collected at days 19, 27, 40 and 51. The percentage of mice that developed measurable serum antibody responses against C1-RBD-C-tag antigen was calculated. At day 19, no measurable titers for mice immunized with 20 μg RBD-C-tag antigen adjuvanted with 320 μg Alum, 50% responders for 500 μg Alum adjuvanted and 60% responders for 650 μg Alum adjuvanted were determined. At day 27, 50% of the mice that were immunized with 20 μg RBD-C-tag antigen adjuvanted with 320 μg Alum developed measurable titers, 80% of the mice that were immunized with for 500 μg Alum adjuvanted and 60% responders for 650 μg Alum adjuvanted were determined. At day 40, 70% of the mice that were immunized with 20 μg RBD-C-tag antigen adjuvanted with 320 μg Alum developed measurable titers, 100% of the mice that were immunized with for 500 μg Alum adjuvanted and 100% responders for 650 μg Alum adjuvanted were determined. At day 51 after Prime, Boost, Boost, 100% of mice immunized developed measurable serum antibody responses against C1-RBD-C-tag antigen.

**Figure 7 vaccines-10-02119-f007:**
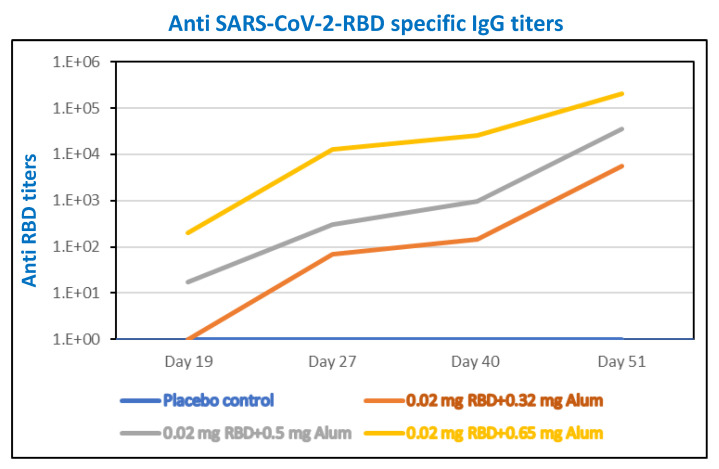
Serum antibody responses against C1-RBD-C-tag antigen in RBD-immunized BALB/c mice. Reciprocal end-point titers of anti-RBD-C-tag immunoglobulin antibody in sera from BALB/c mice immunized subcutaneously with 20 μg RBD-C-tag antigen per mouse, adjuvanted with either 320 μg, 500 μg, or 650 μg Alhydrogel^®^‘85’ representing Alhydrogel^®^‘85’/RBD-C-tag, adjuvant:antigen ratios of 16:1, 25:1 and 32:1 (*w*/*w*), respectively. Titer values of peripheral blood sera were collected at days 19, 27, 40 and 51.

**Figure 8 vaccines-10-02119-f008:**
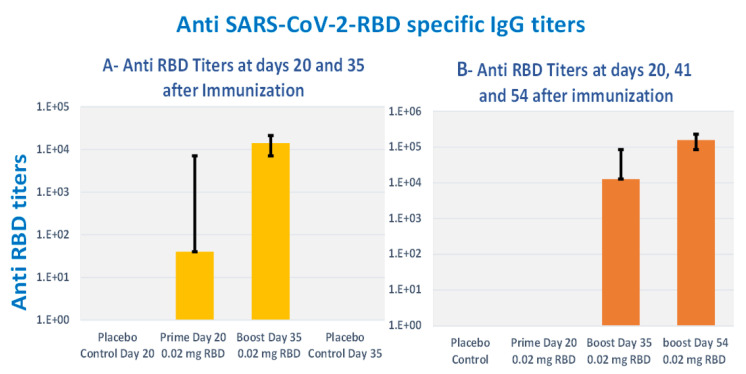
Serum antibody responses generated by Alhydrogel^®^‘85’ adjuvanted C1-RBD-C-tag antigen in K18-hACE2 transgenic C57BL/6J mice: (**A**) reciprocal end-point titers of anti-RBD-C-tag immunoglobulin antibody in sera from K18-hACE2 C57BL/6J transgenic mice immunized twice, each time subcutaneously with 640 μg of Alhydrogel^®^‘85’ adjuvant and 20 μg RBD-C-tag antigen, 20-days after 1st immunization, Prime and 14-days after the second immunization, Boost; and (**B**) anti-RBD-C-tag immunoglobulin titers of mice immunized three times, 20 days after the first immunization, Prime, 20 days after the second immunization, Boost 1, and 13 days after the third immunization, Boost 2.

**Figure 9 vaccines-10-02119-f009:**
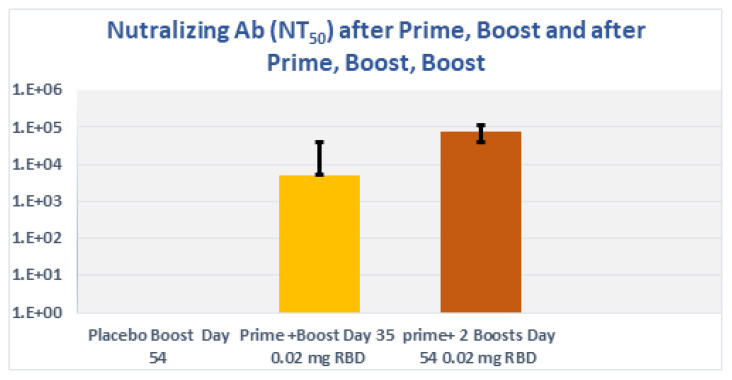
SARS-CoV-2 neutralizing titer generated by Alhydrogel^®^‘85’ adjuvanted C1-RBD-C-tag antigen in K18-hACE2 transgenic C57BL/6J mice. Neutralizing titers determined at day 35, 14 days after the second immunization and neutralizing titers determined at day 54, 13 days after the third immunization. Control mice groups included 640 μg of Alhydrogel^®^‘85’ adjuvant alone without vaccine RBD antigen.

**Figure 10 vaccines-10-02119-f010:**
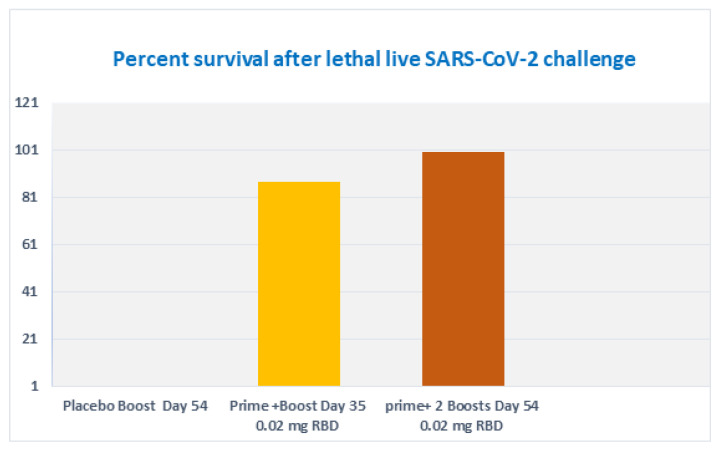
Precent survival of vaccinated K18-hACE2 transgenic C57BL/6J mice after Intra-Nasal challenge with lethal live SARS-CoV-2 virus. The prime-boost immunization regime protects 87.5% of mice that were challenged on day 41, 20 days after the boost immunization. The prime boost, boost immunization regime protected 100% of mice challenged on day 56, 14 days after the second boost immunization from the challenge with lethal live virus. All mice were monitored 21 days after the challenge.

**Figure 11 vaccines-10-02119-f011:**
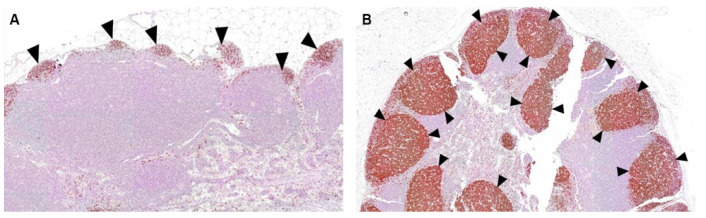
(**A**) The iliac lymph node from an animal injected with the Alhydrogel^®^‘85’, Placebo (Group 1, control item), sacrificed 42 days post first dosing (Recovery phase). Note, arrowheads-no evidence of germinal centers (normal appearance compared with figure b). Staining with antibody for PAX-5 (Biocare’s PAX clone BC/24); and (**B**) the iliac lymph node from an animal injected with C1-RBD Vaccine (Group 2, test item), sacrificed 42 days post first dosing (Recovery phase). The lesions (arrowhead) consist of mild germinal centers increased lymphocytic cellularity (i.e., follicular hyperplasia). Compare with (**A**).

**Table 1 vaccines-10-02119-t001:** Protein subunit SARS-CoV-2 vaccines authorized for use.

Vaccine	Manufacturer	Production Platform	Antigen/ Adjuvant	Current Approvals
ZIFIVAX (ZF2001)	Anhui Zhifei Longcom Biopharmaceutical	CHO K1 cells	RBD-Dimer *Alum **	4 EUA * [30]
CIGB-66, ABDALA	Center for Genetic Engineering and Biotechnology	*Pichia pastoris*	RBD *Alum*	4 EUA [30]
SOBERANA-02	Finlay Institute de Vaccines	CHO cells	RBD conjugated to *Tetanus Toxoid antigen* *Alum*	4 EUA [30]
SOBERANA PLUS	Finlay Institute de Vaccines	CHO cells	RBD-Dimer conjugated to *Tetanus Toxoid antigen* *Alum*	1 EUA [30]
CORBEVAX^TM^	Biological E Limited	*Pichia pastoris*	RBD *CpG 1018* *Alum*	1 EUA [30]
NVX-CoV2373	Novavax	*Sf9* insect-cells	Full-Spike Matrix-M1	EUA [30] 31 countries authorizatios
COVOVAX	Serum Institute of India	*Sf9* insect-cells	Full-Spike *Matrix-M1*	EUA [30] 31 countries authorizations
EpiVaCorona Aurora-CoV	Vektor State Research Center of Virology and Biotechnology	Synthesized	Spike protein oligopeptide *Alum*	4 EUA [30]
EpiVacCorona-N	Vektor State Research Center of Virology and Biotechnology	Synthesized	Spike protein oligopeptide *Alum*	1 EUA [30]
Noora vaccine	Bagheiat-allah University of Medical Sciences	*E. coli* BL21 DE3	RBD *Alum*	1 EUA [30]
V-01	Livzon Mabpharm Inc.	CHO cells	RBD-Fc dimer *Alum*	1 EUA [30]
MVC-COV1901	Medigen	CHO cells	S-2P Spike *CpG 1018* *Alum*	1 EUA [30]
NVSI-06-08	National Vaccine and Serum Institute	CHO cells	trimeric RBD *Alum*	2 EUA [30]
IndoVac	PT Bio Farma	Pichia	RBD *CpG 1018* *Alum*	1 EUA [30]
Razi Cov Pars	Razi Vaccine and Serum Research Institute	Expi293F Cells	Spike *RAS-01*	1 EUA [30]
CoV2 preS dTM D614/B.1.351	Sanofi/GSK	*Sf9* insect-cells	spike trimers *ASO3*	30 EUA [30]
SKYCovione GBP510	SK Bioscience Co., Ltd.	Expi293F Cells	RBD Nanoparticles *ASO3*	1 EUA [30]
TAK-019	Takeda	*Sf9* insect-cells	Full-Spike *Matrix-M1*	1 EUA [30]
SpikoGen, COVAX-19	Vaxine/CinnaGen Co.	*Sf9* insect-cells	Spike *Advax-CpG55.2*	1 EUA [30]

Abbreviations: EUA, Emergency Use Authorization; WHO, World Health Organization. * Alum, Alhydrogel^®^ (Aluminum Hydroxide gel).

## Data Availability

The sequence of the *Trichoderma* reesei, endogenous CBH1 signal sequence, [UniProtKB-P62694 -(GUX1_HYPJE)], cbh1 gene of *Trichoderma* reesei, is available in UniPortKB -https://www.uniprot.org/uniprot/P62694 (accessed on 5 August 2022) and the residues 333-527 of the spike (S1) glycoprotein from SARS-CoV-2 spike S1, strain Wuhan-Hu-1, in (GenBank No.; QHD43416.1)- https://www.ncbi.nlm.nih.gov/protein/1791269090 (accessed on 5 August 2022).

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
