# Peer review of "Thermophilic Filamentous Fungus C1-Cell-Cloned SARS-CoV-2-Spike-RBD-Subunit-Vaccine Adjuvanted with Aldydrogel®85 Protects K18-hACE2 Mice against Lethal Virus Challenge"

_vaccines, 2022, doi:10.3390/vaccines10122119_

Round 1
Reviewer 1 Report
Since constant emergence and spread of SARS-CoV-2 Variants Of Concern (VOC, i.e., Alpha, Beta, Gamma, Delta, Omicron), SARS-CoV-2 perhaps accompany people around the world for nearly three years. This study developed a SARS-CoV-2-Spike-RBD-Subunit-Vaccine Adjuvanted with Aldydrogel®85 based on the genetically modified thermophilic filamentous fungus C1 clonal cell line in order to reduce the production cost for wider use in the future, especially in the lower-income countries. This vaccine candidate demonstrates strong immunogenicity, and antiviral efficacy, including in vivo protection against SARS-CoV-2 challenge in human ACE2-transgenic mice, in view of no loss of body weight or adverse events occurred. Also, it showed no adverse and toxic effects to immunized mouses. According to this study, it provided an effective recombinant protein subunit vaccine with lower production cost, which makes sense in some ways.
Major remarks
1. This introduction is too long to help readers emphasize the importance of this study. Arranging those progress in vaccines for SARS-CoV-2 into a table. Instead, the introduction may be better focus on the meaning and innovation of this study.
2. There are too many long sentences with more than three lines in the whole text. Also, some parts in this study can be merged such as "2.5" and "2.6".
3. Please state some progress about the vaccines based on thermophilic filamentous fungus C1 clonal cell.
4. Please provide flow chart and plasmid profile in the section of plasmid construction.
5. In Fig.8, the day of antibody titers of three injections seemed not at day 35 (day 41).
6. In Fig.9, please complete the legend of the axis in Fig 9B. Also, why did the antibody titer on day 20 in two groups (two and three injections) show great disparity? Did mice in groups with three injections product nearly no antibody on day 20 after prime vaccination?
7. Based on the current situation of COVID-19 infection, more and more people are asymptomatic or with slight symptom. I wonder if it was suitable to challenge with 2,000 PFU (Plaque Forming Units) of lethal live SARS-CoV-2 virus and judge the immune protection according to mortality.
8. Please some more data to prove the safety of this vaccine candidate.
9. It would be better to compare the protection efficacity of this vaccine candidate with other protein subunit vaccines.
Author Response
Dear reviewer her are my corrections and explanations to your comments
Attached is the manuscript after modification requested by Reviewer 1 and 2.
- This introduction is too long to help readers emphasize the importance of this study. Arranging those progress in vaccines for SARS-CoV-2 into a table. Instead, the introduction may be better focus on the meaning and innovation of this study. Introduction modified as requested
- There are too many long sentences with more than three lines in the whole text. Also, some parts in this study can be merged such as "2.5" and "2.6". Modified the best I can
- Please state some progress about the vaccines based on thermophilic filamentous fungus C1 clonal cell. – As sited in References – At the moment there aren’t any vaccines based on thermophilic filamentous fungus C1 clonal cell that were approved for human or animal use. All previous work done with C1 produced antigens was at animal experiments research level. See reverences of this manuscript.
- Please provide flow chart and plasmid profile in the section of plasmid construction. Ok added - Provided as requested Fig 1.
- In Fig.8, the day of antibody titers of three injections seemed not at day 35 (day 41). Correction done in Fig 8 (New 9), in Fig 8 (New 9) legend and in text.
- In Fig.9, please complete the legend of the axis in Fig 9B. Also, why did the antibody titer on day 20 in two groups (two and three injections) show great disparity? Did mice in groups with three injections product nearly no antibody on day 20 after prime vaccination?
Fig 9 (New 10) legend axis completed days of Bleeding added.
As for the difference between the groups. The results were reported as monitored during the experiment. In group A, 4 out of 8 mice responded and had measurable titers after the Prime. In group B only 1 mouse responded.
- Based on the current situation of COVID-19 infection, more and more people are asymptomatic or with slight symptom. I wonder if it was suitable to challenge with 2,000 PFU (Plaque Forming Units) of lethal live SARS-CoV-2 virus and judge the immune protection according to mortality.
The Vaccine candidate was developed for Human use. Recently, the vaccine Receives Regulatory Approval to Initiate Phase 1 Clinical Trial to Demonstrate Clinical Safety and Efficacy in Humans. At the stage of development, we were interested in looking at severe model of immune protection. We were interested in the establishment of the ability of the vaccine to protect naïve and uninfected animals from high dose of lethal live virus infection.
By 1. Antibodies
- Neutralizing AB and Challenge
No loss of body weight or adverse events occurred. DYAI-100A85 also demonstrates excellent safety
- Please some more data to prove the safety of this vaccine candidate.
Please see reference # 71 for GLP-Toxicological study done with DAYI-100
Ramot, Y., Kronfeld, N., Ophir, Y., Ezov, N., Friedman, S., Saloheimo, M., Vitikainen, M., Ben-Artzi, H., Avigdor, A., Tchelet, R., Valbuena, N., Emalfarb, M., Nyska, A. Toxicity and local tolerance of a noval spike protein RBD vaccine against SARS-CoV-2, produced using the C1 Thermothelomyces heterothallica protein platform. Toxicol. Pathol, 50 (3) (2022). https://doi.org/10.1177/01926233221090518
- It would be better to compare the protection efficacity of this vaccine candidate with other protein subunit vaccines.
The protection of the Vaccine against lethal challenge of SARS-CoV-2 was 100% and all other correlates (anti RBD Ab and neutralizing Ab) were comparable to published data sited in the manuscript.
Yakir Ophir

Reviewer 2 Report
In this well-designed study, a relatively novel strategy is tested to attempt to respond to the need for an inexpensive, commercial-scalable, safe and efficacious vaccine for SARS-CoV-2. The investigators have developed a spike receptor-binding domain subunit that is secreted in high quantities by a genetically modified thermophilic filamentous fungus. The authors present a very strong argument for the efficacy of their vaccine candidate. Really, there is nothing to indicate that the vaccine would not be successful. It appears to be easily produced in relatively high yields and is quite easily transported. It elicits a high titer neutralizing and protective response in animals, likely justifying it for human trials.
All of the above is good. However, the presentation in this manuscript is very much overblown, which undoubtedly detracts from its message. When all is said and done, this is really a relatively simple and straightforward study with clearcut results. However, significant modification of the manuscript layout and presentation of the data are badly needed, in order to both enhance its readability and make its message clearer. As written, the manuscript can only be described as bloated
Suggested changes:
1) Page 3, paragraph3: The points made in this paragraph are very outdated, referring to early in 2022. Please update.
2) First paragraph of Materials and Methods: A figure illustrating the design of the expression vector would be helpful here.
3) Table 1 can easily be converted to Supplemental Material.
4) Materials and Methods: At almost ten pages, this section is far too long and needs to be trimmed considerably. Prime candidates for trimming include sections 2.3.1, 2.4, 2.9 2.10 and 2.11.
5) Page 17: It is curious that the authors chose not to show the data for the binding of the RBD-C tag antigen to recombinant hACE2, especially when so much other much less important data are included. These data should be added to the manuscript.
6) The data in Figures 5 and 6 can each be combined into a single panel for each figure.
7) Fig. 7 can easily be eliminated and the data in it presented in the text or at least a small Table.
8) Fig. 8 is clearly the most egregious waste of space and should without question be eliminated. Not only are both points in the figure at 100%, but also the bar is set so low that the data are virtually meaningless. The percent that develop measurable serum antibody responses doesn’t tell us much at all.
Author Response
Dear reviewer her are my corrections and explanations to your comments
Attached is the manuscript after modification requested by Reviewer 1 and 2.
- Page 3, paragraph3: The points made in this paragraph are very outdated, referring to early in 2022. Please update.
Updated and Table 1. Added as requested by Reviewer # 1.
- First paragraph of Materials and Methods: A figure illustrating the design of the expression vector would be helpful here.
A figure illustrating expression vector- added
- Table 1 can easily be converted to Supplemental Material.
Converted to Supplemental Material
- Materials and Methods: At almost ten pages, this section is far too long and needs to be trimmed considerably. Prime candidates for trimming include sections 2.3.1,
2.3.1-Was cut out
As for 2.4- It is the only section in this manuscript that describe the unique production system – In addition section 3.1 in Results is referring to the analysis of the production process.
2.9, 2.10 – Were transferred into Supplemental Material
As for- 2.11 - SARS-CoV-2 virion specific antibody monitored by Plaque Reduction Neutralization Test (PRNT)- This assay is necessary since it is not a standard assay, it is unique for IIBR, especially that SARS-CoV-2 and “home constructed” rVSV-SARS-CoV-2-S were used.
- Page 17: It is curious that the authors chose not to show the data for the binding of the RBD-C tag antigen to recombinant hACE2, especially when so much other much less important data are included. These data should be added to the manuscript
A Fig of a representative binding curve of C1-RBD-C-tag to recombinant hACE2 was added in Supplemental Material
- The data in Figures 5 and 6 can each be combined into a single panel for each figure.
- 7 can easily be eliminated and the data in it presented in the text or at least a small Table. Fig 7 was eliminated and the data in it presented in the text – Section 3.7 Page 18.
Fig. 8 is clearly the most egregious waste of space and should without question be eliminated. Not only are both points in the figure at 100%, but also the bar is set so low that the data are virtually meaningless. The percent that develop measurable serum antibody responses doesn’t tell us much at all.
- Fig 8 was eliminated and the data in it presented in the text – Section 3.8.

Round 2
Reviewer 1 Report
All the questions and suggestions for the manuscript have been answered and revised. I hope this kind of vaccine can be used for people in middle or especially low-income countries with low cost.
Author Response
To reviewer # 1.
Thank you very much for your helpful and constructive comments and suggestions that allowed this manuscript to be published in the Vaccines Journal.
English language and style are fine/minor spell check required.
As requested, I reviewed the manuscript and did minor spell corrections.
Yakir Ophir.
Reviewer 2 Report
This is a revision of a previously reviewed manuscript that explores a relatively novel strategy to generate and characterize an inexpensive, commercially-scalable, safe and effective vaccine for SARS-CoV-2 based on a genetically modified thermophilic filamentous fungus that secretes a spike RBD subunit. The vaccine checks all the boxes for vaccine safety, efficacy, ease of production and transport. Indeed, the authors add the development that the vaccine has recently received regulatory approval for Phase I clinical trials to assess its safety and efficacy in humans.
The authors have been very responsive to the criticism of the previous version with a single exception. They have not addressed the suggestion that the data in the four panels in Figures 5 and 6 can each be combined into a single panel in each figure. Unless, the authors can present a strong case for leaving these two figures as they are, this change should be made. The primary criticism of the original version was that it was excessively long and bloated. However, even after the implementation of most of the other suggestions, the manuscript is pretty much the same length after revision. This change would help a bit. Once this sole criticism is addressed, the manuscript is considered acceptable for publication.
Author Response
To reviewer # 2.
Thank you very much for your helpful and constructive comments and suggestions that allowed this manuscript to be published in the Vaccines Journal.
“The authors have been very responsive to the criticism of the previous version with a single exception. They have not addressed the suggestion that the data in the four panels in Figures 5 and 6 can each be combined into a single panel in each figure.
Once this sole criticism is addressed, the manuscript is considered acceptable for publication”
As requested, Figures 5 and 6 (Figures 6 and 7 of the revised manuscript) were each combined into a signal panel.
Yakir Ophir.